# Quantum Speedups of Optimizing Approximately Convex Functions with Applications to Logarithmic Regret Stochastic Convex Bandits

**Tongyang Li**

Advanced Institute of Information Technology,
Center on Frontiers of Computing Studies,
and School of Computer Science,
Peking University
`tongyangli@pku.edu.cn`

**Ruizhe Zhang**

Department of Computer Science
The University of Texas at Austin
`ruizhe@utexas.edu`

## Abstract

We initiate the study of quantum algorithms for optimizing approximately convex functions. Given a convex set $\mathcal{K} \subseteq \mathbb{R}^n$ and a function $F: \mathbb{R}^n \to \mathbb{R}$ such that there exists a convex function $f: \mathcal{K} \to \mathbb{R}$ satisfying $\sup_{x \in \mathcal{K}} |F(x) - f(x)| \leq \epsilon/n$, our quantum algorithm finds an $x^* \in \mathcal{K}$ such that $F(x^*) - \min_{x \in \mathcal{K}} F(x) \leq \epsilon$ using $\tilde{O}(n^3)$ quantum evaluation queries to $F$. This achieves a polynomial quantum speedup compared to the best-known classical algorithms. As an application, we give a quantum algorithm for zeroth-order stochastic convex bandits with $\tilde{O}(n^5 \log^2 T)$ regret, an exponential speedup in $T$ compared to the classical $\Omega(\sqrt{T})$ lower bound. Technically, we achieve quantum speedup in $n$ by exploiting a quantum framework of simulated annealing and adopting a quantum version of the hit-and-run walk. Our speedup in $T$ for zeroth-order stochastic convex bandits is due to a quadratic quantum speedup in multiplicative error of mean estimation.

## 1 Introduction

Optimization theory is a central research topic in computer science, mathematics, operations research, etc. Currently, many efficient algorithms for optimizing convex functions have been proposed (see for instance [10]), but much less is known for nonconvex optimization. In this paper, we investigate polynomial-time algorithms for optimizing approximately convex functions. On the one hand, such algorithms enjoy robustness and cover many natural scenarios including stochastic convex optimization, empirical risk minimization, etc. On the other hand, approximately convex optimization paves the way of understanding nonconvex optimization in the general case.

Specifically, let $\mathcal{K} \subseteq \mathbb{R}^n$ be a convex set. We call $F: \mathbb{R}^n \to \mathbb{R}$ as an *approximately convex function* over $\mathcal{K}$ if there is a convex function $f: \mathcal{K} \to \mathbb{R}$ such that

$$\sup_{x \in \mathcal{K}} |F(x) - f(x)| \leq \epsilon/n. \tag{1}$$

Throughout the paper, we assume that $f$ is $L$-Lipschitz with respect to $\ell_\infty$ norm, i.e., $|f(x) - f(y)| \leq L\|x - y\|_\infty$ for any $x, y \in \mathcal{K}$. We assume that $\mathcal{B}_2(0, 1) \subseteq \mathcal{K} \subseteq \mathcal{B}_2(0, R)$, i.e., the convex body $\mathcal{K}$ contains the unit ball centered at 0 and is contained by a ball of radius $R$ centered at 0. Unless otherwise mentioned, we aim at algorithms with $\mathrm{poly}(\log R)$ dependence in $R$.

It is standard to assume the *zeroth-order oracle* of $F$, which returns the function value $F(x)$ given an input $x$. As far as we know, the state-of-the-art algorithm for finding an $x^* \in \mathcal{K}$ such that $F(x^*) -$

36th Conference on Neural Information Processing Systems (NeurIPS 2022).

$\min_{x \in \mathcal{K}} F(x) \leq \epsilon$ with high probability was proposed by Belloni et al. [8], which takes $\tilde{O}(n^{4.5})$[1] queries to the zeroth-order oracle. On the other hand, Ref. [34] proved that if the approximation error in (1) is at least $\tilde{\Omega}(\max\{\epsilon/n, \epsilon^2/\sqrt{n}\})$, there exists a function $F$ which no algorithm can find a point $x^* \in \mathcal{K}$ such that $F(x^*) - \min_{x \in \mathcal{K}} F(x) \leq \epsilon$ using $\text{poly}(n, 1/\epsilon)$ queries to $F$. In other words, the $\epsilon/n$ term in (1) is fundamental, and it has been the standard assumption of studying approximately convex optimization in [8, 34].

Based on approximate convex functions, a closely related scenario is *stochastic convex functions*, where we have a function $F \colon \mathbb{R}^n \to \mathbb{R}$ such that

$$F(x) = f(x) + \epsilon_x.$$

Here $f \colon \mathbb{R}^n \to \mathbb{R}$ is a convex function and $\epsilon_x$ is a sub-Gaussian random variable with parameter $\sigma$, i.e., $\mathbb{E}[\exp(\lambda \epsilon_x)] \leq \exp(\sigma^2 \lambda^2/2)$ for any $\lambda$. This implies that

$$\Pr[|\epsilon_x| \geq \sigma t] \leq 2 \exp(-t^2/2) \quad \forall t \geq 0. \tag{2}$$

Using this fact, Belloni et al. [8] gave an algorithm for stochastic convex optimization with $\tilde{O}(n^{7.5}/\epsilon^2)$ queries to a zeroth-order oracle of $F$.

**Contributions.** In this paper, we conduct a systemic study of quantum algorithms for optimization of approximately convex functions, with applications to zeroth-order stochastic convex bandits. Quantum computing is a rapidly advancing technology, the capability of quantum computers is dramatically increasing and recently reached "quantum supremacy" by Google [6] and USTC [42]. In optimization theory, quantum advantages have been proven for semidefinite programs [3, 4, 11, 12], general convex optimization [5, 15], the escaping from saddle point problem in optimization [41], etc. Nevertheless, as far as we know, quantum algorithms for approximately convex optimization and stochastic convex optimization are widely open. In this paper, we consider these problems using the quantum zeroth-order evaluation oracle $O_F$, a standard model used in previous quantum computing literature [5, 14, 15, 41]:

$$O_F |x, y\rangle = |x, F(x) + y\rangle \quad \forall x \in \mathbb{R}^n, y \in \mathbb{R}. \tag{3}$$

Here $|\cdot\rangle$ is the Dirac notation, and preliminaries of quantum computing will be covered in Section 2. Intuitively, Eq. (3) can take inputs with form $\sum_{i=1}^m \mathbf{c}_i |\mathbf{x}_i\rangle \otimes |0\rangle$ where $\mathbf{x}_i \in \mathbb{R}^n \, \forall i \in [m]$ and $\sum_{i=1}^m |\mathbf{c}_i|^2 = 1$, and if we measure the outcome quantum state, we get $F(\mathbf{x}_i)$ with probability $|\mathbf{c}_i|^2$. In particular, the quantum zeroth-order oracle allows the ability to query different locations in *superposition*, which is stronger than the classical counterpart (i.e., $m = 1$). Nevertheless, if the classical zeroth-order oracle can be implemented by explicit arithmetic circuits, the quantum oracle in (3) can be implemented by quantum circuits of the same size up to logarithmic factors. As for stochastic convex functions, similar to (3), we assume the following oracle:

$$O_f |x, y\rangle = |x\rangle \int_{\xi \in \mathbb{R}} \sqrt{g_x(\xi)} |f(x) + y + \xi\rangle \, \mathrm{d}\xi \quad \forall x \in \mathbb{R}^n, y \in \mathbb{R}, \tag{4}$$

where for any $x \in \mathbb{R}^n$, $g_x(\xi)$ follows a sub-Gaussian distribution as in (2).

Our first result is a quantum algorithm for optimizing approximately convex functions:

**Theorem 1.** *With probability at least* 0.9*, we can find an $x^* \in \mathcal{K}$ such that*

$$F(x^*) - \min_{x \in \mathcal{K}} F(x) \leq \epsilon \tag{5}$$

*using $\tilde{O}(n^3)$ queries to the quantum evaluation oracle (3).*

We remark that the succss probability can be easily boosted up to $1 - \delta$ for any $\delta \in (0, 1)$, by paying an extra $\log(1/\delta)$ factor in the quantum query complexity. Compared to the best-known classical result by Belloni et al. [8] with query complexity $\tilde{O}(n^{4.5})$, we achieve a polynomial quantum speedup in terms of $n$. Technically, Belloni et al.'s algorithm is based on the simulated annealing process and using the Hit-and-Run walk to generate samples in each stage. In this work, we give a *user-friendly*

---

[1]The $\tilde{O}$ and $\tilde{\Omega}$ notation omits poly-logarithmic terms, i.e., $\tilde{O}(g) = O(g \operatorname{poly}(\log g))$ and $\tilde{\Omega}(g) = \Omega(g \operatorname{poly}(\log g))$. Unless otherwise mentioned, both notations also omit $\operatorname{poly}(\log 1/\epsilon)$ terms.

version of the quantum walk framework (Theorem 3) that can be applied very easily to obtain quantum speedup for the mixing of classical Markov chains. We then analyze the warmness and the overlap between adjacent Hit-and-Run walks in the simulated annealing process, showing that our quantum walk framework is applicable to each classical Hit-and-Run sampler. By implementing the random walk quantumly, we improve the query complexity of the sampling procedure from $\widetilde{O}(n^3)$ to $\widetilde{O}(n^{1.5})$ (Theorem 4). We also design a non-destructive rounding procedure (Lemma 4) in each stage of the simulated annealing that simulates the classical rounding procedure in [8] but will not destroy the quantum states of the quantum walks. Combining them together gives the $\widetilde{O}(n^3)$-query quantum algorithm for minimizing an approximately convex function. Our result can also be applied to give an $\widetilde{O}(n^3)$-query quantum algorithm for optimizing approximately convex functions with decreasing fluctuations.

See Section 3 for more details and the proof of Theorem 1.

Furthermore, to estimate the mean of a random variable up to multiplicative error $\epsilon$, classical algorithms need to take $1/\epsilon^2$ samples due to concentration inequalities such as Chernoff's bound, whereas quantum algorithms can take roughly $1/\epsilon$ queries (see Proposition 1). As a result, we obtain polynomial quantum speedup in both $n$ and $1/\epsilon$ for stochastic convex optimization:

**Corollary 1.** *With probability at least* $0.8$*, we can find an* $x^* \in \mathcal{K}$ *such that*

$$f(x^*) - \min_{x \in \mathcal{K}} f(x) \leq \epsilon \tag{6}$$

*using* $\tilde{O}(n^5/\epsilon)$ *queries to the quantum stochastic evaluation oracle (4).*

We apply Corollary 1 to solve the zeroth-order stochastic convex bandit problem, which is a widely studied bandit model (see e.g., [22, 24, 25]). The problem is defined as follows. Let $\mathcal{K} \subseteq \mathbb{R}^n$ be a convex body and $f \colon \mathcal{K} \to [0, 1]$ be a convex function. Here $\mathcal{B}_2(0, 1) \subseteq \mathcal{K} \subseteq \mathcal{B}_2(0, R)$. An online learner and environment interact alternatively over $T$ rounds. In each round $t \in [T]$, the learner makes a query to the quantum stochastic evaluation oracle (4), and returns a value $x_t \in \mathcal{K}$ as the current guess. The learner aims to minimize the regret

$$\mathcal{R}_T := \mathbb{E}\left[\sum_{t=1}^{T}(f(x_t) - f^*)\right], \quad \text{where } f^* = \min_{x \in \mathcal{K}} f(x), \tag{7}$$

and the expectation is taken over all randomness. The classical state-of-the-art algorithm using a classical stochastic evaluation in each round achieves a regret bound of $\tilde{O}(n^{4.5}\sqrt{T})$ [25], and there is a classical lower bound $\Omega(n\sqrt{T})$ on the regret [19]. Here we prove:

**Theorem 2.** *There is a quantum algorithm for which* $\mathcal{R}_T = \tilde{O}(n^5 \log(T) \log(TR))$*.*

This achieves $\text{poly}(\log T)$ regret for zeroth-order stochastic convex bandits, **an exponential quantum advantage** in terms of $T$ compared to classical zeroth-order stochastic convex bandits. As far as we know, we give the first quantum algorithms with poly-logarithmic regret bound on online learning problems. An independent work [37] gave quantum algorithms with poly-logarithmic regret for multi-armed bandits and stochastic linear bandits, but these two types of bandits concern reward of discrete objects and linear functions, respectively, which are fundamentally different from the stochastic convex bandits we study.

To achieve this $\text{poly}(\log T)$ regret for zeroth-order stochastic convex bandits, we divide the $T$ iterations into $\lfloor \log_2 T \rfloor$ intervals with doubling length $1, 2, 4, \ldots$. We use the quantum queries from a previous interval to run our quantum stochastic convex optimization algorithm and use the output as the guess for all iterations in the next interval. Since we can achieve linear dependence in $1/\epsilon$ in Corollary 1, it can be calculated that the total regret in each iteration is at most $\tilde{O}(n^5 \log(TR))$, leading to our claim. The proof details are given in Section 4.

**Open questions.**    Our work raises several natural questions for future investigation:

- Can we further improve the dimension dependence of our approximate convex optimization algorithm? The current quantum speedup mainly leverages the quantum hit-and-run walk; it is of general interest to understand whether we can gain quantum advantage at other steps, or whether the convergence analysis of the hit-and-run walk per se can be improved.

- Can we improve the dimension dependence of the regret of our zeroth-order stochastic convex bandits? It is natural to check whether our $n^5$ term can be improved by quantizing the classical state-of-the-art [25] and other recent works.
- Can we give fast quantum algorithms for more general nonconvex optimization problems, or with poly-logarithmic regret for more general bandit problems?

**Limitations and societal impacts.** This work is purely theoretical. Researchers working on theoretical aspects of quantum computing and optimization theory may benefit from our results at the moment. In the long term, once we have fault-tolerant quantum computers, our results can be applied to approximate convex optimization and stochastic bandit scenarios arising in the real world. As far as we are aware, our work does not have negative societal impacts.

## 2 Preliminaries

### 2.1 Basics of quantum computing

We briefly introduce basic notations and concepts of quantum computing in this section. More details are covered in standard textbooks, for instance [30].

Quantum computing can be formulated in terms of linear algebra. Specifically, the computational basis of space $\mathbb{C}^d$ can be denoted by $\{\vec{e}_1, \ldots, \vec{e}_d\}$, where $\vec{e}_i = (0, \ldots, 1, \ldots, 0)^\top$ with the $i^{\text{th}}$ entry being 1 and other entries being 0. These basis vectors are typically written by the *Dirac notation*, where we denote $\vec{e}_i$ by $|i\rangle$ (called a "ket"), and denote $\vec{e}_i^\top$ by $\langle i|$ (called a "bra").

A $d$-dimensional *quantum state* is an $\ell_2$-norm unit vector in $\mathbb{C}^d$, e.g., $|v\rangle = (v_1, \ldots, v_d)^\top$ such that $\sum_i |v_i|^2 = 1$. For each $i$, $v_i$ is called the *amplitude* in $|i\rangle$. The *tensor product* of quantum states is their Kronecker product: if $|u\rangle \in \mathbb{C}^{d_1}$ and $|v\rangle \in \mathbb{C}^{d_2}$, then

$$|u\rangle \otimes |v\rangle := (u_1 v_1, u_1 v_2, \ldots, u_{d_1} v_{d_2})^\top \in \mathbb{C}^{d_1} \otimes \mathbb{C}^{d_2}.$$

We often omit the operator $\otimes$ and simply write $|u\rangle |v\rangle$ or $|u, v\rangle$ for being concise.

In general, the definition of quantum states can be extended to a continuous domain. For instance,

$$|v\rangle = \int_{\mathbb{R}^n} v_x |x\rangle \mathrm{d}x$$

represents a quantum state as long as $\int_{\mathbb{R}^n} |v_x|^2 \mathrm{d}x = 1$. To keep quantum states normalized in $\ell_2$ norm, operations in quantum computing are *unitary transformations*.

### 2.2 Classical and quantum walks

A classical Markov over the space $\Omega$ is a sequence of random variables $\{X_i\}_{i \in \mathbb{N}}$ such that for any $i > 0$ and $x_0, \ldots, x_i \in \Omega$,

$$\Pr[X_i = x_i \mid X_0 = x_0, \cdots, X_{i-1} = x_{i-1}] = \Pr[X_i = x_i \mid X_{i-1} = x_{i-1}].$$

The Markov chain can be represented by its stochastic transition matrix $P$ such that $\sum_{y \in \Omega} P(x, y) = 1$ for any $x \in \Omega$. A distribution $\pi$ is *stationary* if it satisfies $\sum_{x \in \Omega} \pi(x) P(x, y) = \pi(y)$ for any $y \in \Omega$. A Markov chain is *reversible* if its stationary distribution satisfies the following detailed balance condition:

$$\pi(x) P(x, y) = \pi(y) P(y, x) \quad \forall x, y \in \Omega.$$

The *mixing time* of a Markov chain with initial distribution $\pi_0$ is the number of steps $t = t(\epsilon) \in \mathbb{N}$ such that the total variation distance between the time-$t$ distribution and the stationary distribution is at most $\epsilon$ for any $\epsilon \in (0, 1)$, i.e.,

$$d_{\text{TV}}(P^t \pi_0, \pi) \leq \epsilon.$$

In quantum, we can also define the discrete-time quantum walk [36], which is a quantum-analogue of classical Markov chain. More precisely, we use a quantum state to represent a classical probability distribution:

$$\{\pi(x)\}_{x \in \Omega} \longleftrightarrow |\pi\rangle = \sum_{x \in \Omega} \sqrt{\pi(x)} |x\rangle.$$

In quantum walk, there are two operations:

- **Reflect:** An operator $R$ that reflects the quantum states with respect to the subspace $\text{span}\{|x\rangle |\psi_x\rangle\}_{x\in\Omega}$, where $|\psi_x\rangle = \sum_{y\in\Omega} \sqrt{P(x,y)} |y\rangle$.

- **Swap:** An operator $S$ that swap the two quantum registers: $S |x\rangle |y\rangle = |y\rangle |x\rangle$.

Then, the quantum walk operator $W$ is defined as: $W := S \circ R$. Intuitively, the first quantum register contains the current position of the random walk, and the second register contains the previous position. In each step of quantum walk, the $R$ operator makes a superposition in the second register of the next step positions with amplitudes proportional to their transition probabilities. Then, the $S$ operator swaps the two quantum registers, using the first register to store the new positions and the second register to store the old position. In this way, we complete one-step of the random walk coherently (in superposition).

The quantum advantage of the quantum walk comes from the spectrum of $W$. Note that the state of stationary distribution $\sum_{x\in\Omega} \sqrt{\pi(x)} |x\rangle |\psi_x\rangle$ is invariant under $W$. In other words, it is an eigenvector with eigenvalue 1 (eigenphase 0). On the other hand, the other eigenvectors of $W$ has eigenphase at least $\sqrt{2\delta}$, where $\delta$ is the spectral gap[2] of the transition matrix $P$. Therefore, applying the *quantum phase estimation* algorithm using $O(1/\sqrt{\delta})$ calls to $W$ can distinguish the state corresponding to the stationary distribution and other eigenstates. See [17, 36, 38] for more details.

## 2.3 Hit-and-Run walk

Hit-and-Run walk was introduced by R.L. Smith [35], and has a long line of reserach [1, 7, 9, 16, 26–29, 40] for its mixing time and applications in sampling, optimization, and volume estimation. Intuitively, the Hit-and-Run walk is defined as follows:

1. Pick a uniformly distributed random line $\ell$ through the current point $x$.

2. Move to a random point $y$ along the line chosen from the restricted distribution $\pi_\ell$.

More specifically, let $f : \mathbb{R}^n \to \mathbb{R}$ be a logconcave distribution density. Then, the distribution induced by restricting $f$ to the line $\ell$ is defined as follows:

$$\pi_\ell(S) := \frac{\int_S f(x)\mathrm{d}x}{\int_\ell f(x)\mathrm{d}x} \quad \forall S \subset \ell.$$

The following lemma show the transition probability of the Hit-and-Run walk.

**Lemma 1** ([26]). *Let $f$ be the density of a logconcave distribution. If the current point of Hit-and-Run is $u$, then the density function of the distribution of the next point $x$ is*

$$f_u(x) = \frac{2}{n\pi_n} \frac{f(x)}{\mu_f(u,x)\|x-u\|^{n-1}},$$

*where $\pi_n = \frac{\pi^{n/2}}{\Gamma(1+n/2)}$ and $\mu_f(u,x)$ is the $f$-measure of the chord through $u$ and $x$.*

# 3 Quantum Algorithm for Optimizing Approximately Convex Functions

We give a polynomial quantum speedup for minimizing an approximately convex functions using the quantum walk algorithm [36, 38]. More specifically, we consider the quantum walk in continuous space.[3] Let $P$ denote the (column) stochastic transition density of a reversible Markov chain, i.e.,

$$\int_{\mathbb{R}^n} P(x,y)\mathrm{d}y = 1 \quad \forall x \in \mathbb{R}^n,$$

and let $\pi(x)$ denote the density of the stationary distribution. Then, we can implement a quantum walk unitary $W(P)$ such that its unique eigenvector with eigenvalue 1 (or equivalently eigenphase 0) is:

$$\int_{\mathbb{R}^n} \sqrt{\pi(x)} |x\rangle \otimes |\psi_x\rangle \, \mathrm{d}x,$$

---

[2]The difference between the first and the second largest eigenvalues.

[3]It can be naturally discretized as we do in simulating a Markov chain on classical digital computers.

where $|\psi_x\rangle := \int_{\mathbb{R}^n} \sqrt{P(x,y)} |y\rangle \,\mathrm{d}y$ mixes all the points that can be moved from $x$, with amplitudes proportional to the transition probabilities. This quantum state can be considered as a *coherent encoding* of the classical distribution $\pi$. The advantage of quantum walk comes from the fact that $W(P)$ has phase gap $\sqrt{\delta}$, where $\delta$ is the spectral gap of $P$. Therefore, by the quantum phase estimation algorithm [23], quantum walk can achieve quadratic speedup in $\delta$. In general, quantum walk algorithm can quadratically speedup the hitting time of classical Markov chain. For the mixing time, it requires some additional complicated constraints on the Markov chain and distributions (see e.g., [2, 13, 31, 39]). Based on previous studies on quantum walk mixing, we propose the following user-friendly quantum walk framework:

**Theorem 3** (User-friendly quantum walk framework). *Let $M_0$ be the initial Markov chain with stationary distribution $\pi_0$, $M_1$ be the target Markov chain with stationary distribution $\pi_1$. Suppose $M_0, M_1$ satisfy the following properties:*

- **Mixing time:** $d_{\mathrm{TV}}(P_1^{t_0} \cdot \pi_0, \pi_1) \leq \epsilon$ *and* $d_{\mathrm{TV}}(P_0^{t_1} \cdot \pi_1, \pi_0) \leq \epsilon$.

- **Warmness:** $\|\pi_0/\pi_1\| = O(1)$ *and* $\|\pi_1/\pi_0\| = O(1)$, *where* $\|\pi/\sigma\| := \int_{\mathbb{R}^n} \frac{\pi(x)}{\sigma(x)} \pi(x) \mathrm{d}x$.

- **Overlap:** $|\langle \pi_0 | \pi_1 \rangle| = \int_{\mathbb{R}^n} \sqrt{\pi_0(x)\pi_1(x)} \mathrm{d}x = \Omega(1)$.

*Furthermore, suppose we have access to a unitary $U$ that prepares the initial state $|\pi_0\rangle = \int_{\mathbb{R}^n} \sqrt{\pi_0(x)} |x\rangle \,\mathrm{d}x$. Then, we can obtain a state $|\widetilde{\pi}_1\rangle$ with $\| |\widetilde{\pi}_1\rangle - |\pi_1\rangle \|_2 \leq \epsilon$ using*

$$O\left( \sqrt{t_0 + t_1} \log^2(1/\epsilon) \right)$$

*calls to the quantum walk operators.*

More details and proofs are deferred to Appendix A.

Next, we can use Theorem 3 to speed-up the best-known classical algorithm for optimizing approximately convex functions [8], which has the following three levels:

- **High level:** Perform a simulated annealing with $K$ stages. At the $i$-th stage, the target distribution $\pi_{g_i}$ has density $\propto g_i(x) = e^{-F(x)/T_i}$, where $T_i := (1 - 1/\sqrt{n})^i$.

- **Middle level:** Use $N$ samples from $\pi_{g_i}$ to construct a linear transformation $\Sigma_i$, rounding the distribution to near-isotropic position.

- **Low level:** Run the hit-and-run walk to evolve the distribution from $\pi_{g_{i-1}}$ to $\pi_{g_i}$.

Here, each step of the hit-and-run walk picks a uniformly random direction at the current point, and then walks on the 1-dimensional chord intersected by the direction and $\mathcal{K}$ with probability density proportional to the logconcave density. We formally state the hit-and-run walk in Algorithm 3.

We focus on speeding-up the Low level using Theorem 3. Hence, we need to show that $\pi_{g_i}$ and $\pi_{g_{i+1}}$ satisfy the properties therein. First of all, it has been proved in [8] that $\|\pi_{g_i}/\pi_{g_{i+1}}\| = O(1)$ (Lemma 11). We prove the following lemma with proof deferred to Appendix C.1.

**Lemma 2** (Informal version of Lemma 12). *Let $\pi_{g_i}$ be a distribution with density proportional to $g_i(x) = \exp(-F(x)/T_i)$, where $F(x)$ is $\beta$-approximately convex. Then, for any $0 \leq i \leq K - 1$,*

$$\|\pi_{g_{i+1}}/\pi_{g_i}\| \leq 8\exp(2\beta/T_{i+1}) \leq O(1).$$

Therefore, the warmness property is satisfied.

We also prove that the Markov chains in this annealing schedule are slowly evolving:

**Lemma 3** (Informal version of Lemma 13). *Let $\pi_{g_i}$ be a distribution with density proportional to $g_i(x) = \exp(-F(x)/T_i)$, where $F(x)$ is $\beta$-approximately convex. Then, for any $0 \leq i \leq K - 1$,*

$$|\langle \pi_i | \pi_{i+1} \rangle| \geq \exp(-(\beta/T_{i+1} + 1)/2) = \Omega(1).$$

Hence, the overlap property is also satisfied.

With the warmness and classical analysis of the hit-and-run walk (Theorem 7), we get that the classical mixing time from $\pi_{g_i}$ to $\pi_{g_{i+1}}$ and vice versa can be bounded by $\widetilde{O}(n^3)$.

Moreover, we also show in Appendix C.1 and Lemma 14 that each call to the quantum walk operator can be implemented by querying the evaluation oracle $O(1)$ times.

Thus, by Theorem 3, we get the following theorem:

**Theorem 4** (Low level quantum speedup, informal version of Theorem 11). *Let $\gamma \in (0, 1/e)$. Let $g_i(x) = \exp(-F(x)/T_i)$ be the density of $\pi_{g_i}$ with $F(x)$ being $\beta/2$-approximately convex. Let $T_i = (1 - 1/\sqrt{n})^i$ for $0 \le i \le K$. Then, for each $0 \le i \le K - 1$, given a state $|\pi_{g_i}\rangle$, we can produce a state $|\hat{\sigma}_i^{(m)}\rangle$ such that*

$$\||\pi_{g_{i+1}}\rangle - |\hat{\sigma}_i^{(m)}\rangle\|_2 \le O(\gamma),$$

*using $m = \widetilde{O}(n^{1.5})$ calls for the evaluation oracle of $F$.*

In the Middle level, we need to use $N$ independent samples from $\pi_{g_i}$ to construct a linear transformation $\Sigma_i$ for rounding. However, we cannot directly measure the state $|\pi_{g_i}\rangle$, since it will destroy the quantum coherence. Instead, we propose a non-destructive approach to construct the linear transformation (the proof is deferred to Appendix C.2):

**Lemma 4** (Non-destructive rounding, informal version of Lemma 15). *For each $i \in [K]$, the linear transformation $\Sigma_i$ can be obtained using $\widetilde{O}(N)$ copies of the states $|\pi_{g_{i-1}}\rangle$, with query complexity $\widetilde{O}(N \cdot n^{1.5})$. Moreover, the states $|\pi_{g_{i-1}}\rangle$ will be recovered with high probability.*

Now, we can put all the components together and obtain a quantum algorithm for optimizing approximately convex function with $\widetilde{O}(n^{1.5})$ quantum query complexity (Algorithm 1). We sketch the proof in below and the formal proof is given in Appendix C.3:

*Proof sketch of Theorem 1.* Observe that at each annealing stage, the sample distribution is the same as the classical algorithm. Thus, by the classical analysis (Theorem 9) in [8], the same optimization guarantee still holds for the quantum algorithm. Thus, if we take $K = \sqrt{n}\log(n/\epsilon)$ and $N = \widetilde{O}(n)$, the output $x_*$ of QSIMANNEALING procedure satisfies:

$$F(x_*) - \min_x F(x) \le O(\epsilon)$$

with high probability.

Then, consider the query complexity. There are $K$ stages in the annealing process, where each stage maintains $\widetilde{O}(N)$ samples (quantum states). By Theorem 4, evolving each state takes $\widetilde{O}(n^{1.5})$ queries. Therefore, the total query complexity is

$$K \cdot N \cdot \widetilde{O}(n^{1.5}) = \widetilde{O}(n^3).$$

Here, we assume that the convex body $\mathcal{K}$ is known, e.g., $\mathcal{K} = \mathbb{R}^n$ or $\mathbb{S}^n$. However, even if $\mathcal{K}$ is unknown, we can call its membership oralce to run our algorithm. More specifically, in constructing the initial state $|\pi_0\rangle$, we need to query the membership oracle for $\widetilde{O}(1)$ times. And since we prepare $N$ copies, this step takes $\widetilde{O}(N)$ queries in total. Then, in each step of the quantum walk, we need to query the membership oracle for $\widetilde{O}(1)$ times to determine the intersection point for the hit-and-run process. Thus, the number of queries to the membership oracle of $\mathcal{K}$ is the same as the the number of evaluation oracle queries. Hence, our algorithm will query the membership oracle for $\widetilde{O}(n^3)$ times in all.

As for the number of qubits to implement our algorithm, each state $|\pi_0\rangle$ uses $\widetilde{O}(n)$ qubits. Thus, $\widetilde{O}(n^2)$ qubits are used to store all states. And we need $O(n)$ ancilla qubits for the quantum walk unitaries. Therefore, our algorithm uses $\widetilde{O}(n^2)$ qubits in total. $\qquad\square$

**Optimization of approximately convex functions with decreasing fluctuations.** Beyond the $\ell_\infty$-norm assumption for all $x \in \mathcal{K}$ in Eq. (1), it is also possible to give efficient algorithms for optimizing other types of approximately convex functions. Specifically, Belloni et al. [8, Section 7] studied approximately convex functions with decreasing fluctuations.

---

**Algorithm 1** Quantum speedup for approximately convex optimization (Informal version)

---

1: **procedure** QSIMANNEALING          ▷ Theorem 1
2:     $N \leftarrow \widetilde{O}(n), K \leftarrow \sqrt{n}\log(n/\epsilon)$
3:     Prepare $N$ (approximately) copies of $|\pi_0\rangle$, denoted as $|\widetilde{\pi}_0^{(1)}\rangle, \ldots, |\widetilde{\pi}_0^{(N)}\rangle$
4:     **for** $i \leftarrow 1, \ldots, K$ **do**
5:        Use $\{|\widetilde{\pi}_0^{(j)}\rangle\}_{j\in[N]}$ to nondestructively construct $\Sigma_i$         ▷ Lemma 4
6:        Apply quantum walk to evolve the states $|\widetilde{\pi}_{i-1}^{(j)}\rangle$ to $|\widetilde{\pi}_i^{(j)}\rangle$         ▷ Theorem 4
7:     **end for**
8:     $x_K^j \leftarrow$ measure the final state $|\widetilde{\pi}_K^{(j)}\rangle$ for $j \in [N]$
9:     **return** $\arg\min_{j\in[N]} F(x_K^j)$
10: **end procedure**

---

Suppose that the function $f$ in (1) is 1-Lipschtiz and $\alpha$-strongly convex with minimum at $x_{\min} \in \mathcal{K}$:

$$f(x) - f(x_{\min}) \geq \langle \nabla f(x_{\min}), x - x_{\min}\rangle + \frac{\alpha}{2}\|x - x_{\min}\|^2 \geq \frac{\alpha}{2}\|x - x_{\min}\|^2.$$

Define a measure of "non-convexity" of $F$ w.r.t to $f$ in an $n$-dimensional ball of radius $r$ near $x_{\min}$:

$$\Delta(r) := \sup_{x \in \mathcal{B}_2(x_{\min}, r)} |F(x) - f(x)|.$$

We can call Theorem 1 iteratively. Suppose that at the start of the $t^{\text{th}}$ iteration we have a ball $\mathcal{B}_2(x_{t-1}, 2r_{t-1})$ satisfying

$$\mathcal{B}_2(x_{\min}, r_{t-1}) \subset \mathcal{B}_2(x_{t-1}, 2r_{t-1}) \subset \mathcal{B}_2(x_{\min}, 3r_{t-1}).$$

After executing Theorem 1 in this iteration with $\widetilde{O}(n^3)$ quantum queries, we reach a point $x_t$ such that with high probability

$$f(x_t) - f(x_{\min}) \leq Cn\Delta(3r_{t-1}))$$

for some global constant $C > 0$. Due to strong convexity, this gives a new radius $r_t$ recursively:

$$\frac{\alpha}{2Cn}r_t^2 := \frac{\alpha}{2Cn}\|x_t - x_{\min}\|^2 \leq \Delta(3r_{t-1}).$$

When $\Delta(r) = cr^p$ for some $c > 0, p \in (0, 2)$, the iteration stops when

$$r^* = (2 \cdot 3^p cCn/\alpha)^{1/(2-p)}$$

, and when

$$\Delta(r) = c\log(1 + dr)$$

for some $c, d > 0$, the iteration stops at $r^*$ satisfying

$$(2cCn/\alpha)\log(1 + 3dr^*) = (r^*)^2.$$

The total number of quantum queries is still $\widetilde{O}(n^3)$.

## 4   Quantum Algorithm for Zeroth-Order Stochastic Convex Bandits

We first prove Corollary 1 using quantum mean estimation with a Gaussian tail:

**Proposition 1** (Adapted from [20, Theorem 4.2])**.** *Suppose that $X$ is a random variable on a probability space $(\Omega, p)$ with mean $\mu$ and variance $\sigma^2$. Suppose we have a unitary oracle $U$ satisfying $U|0\rangle = \int_{x\in\Omega} \sqrt{p(x)}|x\rangle \mathrm{d}x$. Then, for any $\Delta \in (0, 1)$ and $\tau \in \mathbb{N}$ such that $\tau \geq \log(1/\Delta)$, there is a quantum algorithm that outputs a mean estimate $\tilde{\mu}$ such that*

$$\Pr\left[|\tilde{\mu} - \mu| > \frac{\sigma\log(1/\Delta)}{\tau}\right] \leq \Delta,$$

*using $O(\tau \log^{3/2}(\tau)\log\log(\tau))$ queries to $U$.*

*Proof of Corollary 1.* We follow Section 6 of [8] while use Theorem 1 and Proposition 1. Specifically, for a parameter $0 < \alpha < 1$, we let $\mathcal{N}_\alpha$ be a box grid of $\mathcal{K}$ with side length $\alpha$. In other words, $\mathcal{N}_\alpha$ is $\alpha$-net of $\mathcal{K}$ in $\ell_\infty$ norm. Since $\mathcal{K} \subseteq \mathcal{B}_2(0, R)$, $|\mathcal{N}_\alpha| \leq (R/\alpha)^n$.

Note that for the sub-Gaussian random variable $\epsilon_x$ in (2), it has variance at most $4\sigma^2$ because

$$\mathbb{E}[|\epsilon_x|^2] = \int_0^\infty \Pr[|\epsilon_x| > \sqrt{s}]\mathrm{d}s \leq 2\int_0^\infty e^{-\frac{s}{2\sigma^2}}\mathrm{d}s = 4\sigma^2.$$

Upon a query $x' \in \mathcal{K}$, we define an oracle $O_f^{\tau,\alpha}$ which returns $f(x) + \tilde{\epsilon}_x$ for $x \in \mathcal{N}_\alpha$ which is closest to $x'$, and the $\tilde{\epsilon}_x$ here is obtained by applying Proposition 1 with the unitary oracle $O_f$ in Eq. (4) to estimate the function value $f(x)$. Specifically, with $\Delta = \exp(-t^2)$ where $t$ is a parameter determined later, we have

$$\Pr\left[|\tilde{\epsilon}_x| > \frac{\sigma t^2}{\tau}\right] \leq \exp(-t^2). \tag{8}$$

We note that in our algorithm based on the hit-and-run walk, with probability 1 we do not revisit the same point. As a result, $O_f^{\tau,\alpha}$ is no more powerful than $O_f$ since the learner only obtains information on $\mathcal{N}_\alpha$, and in the rest of the proof we assume $O_f^{\tau,\alpha}$ as the oracle we use. We take

$$\alpha = \epsilon/2nL, \quad t = \sqrt{n\ln(R/\alpha) + \ln 10}.$$

Note that the value of $t$ promises that $\exp(-t^2)(R/\alpha)^n \leq 0.1$. In other words, with probability at least 0.9, we promise that

$$\max_{x \in \mathcal{N}_\alpha} |\tilde{\epsilon}_x| \leq \frac{\sigma t^2}{\tau} = \frac{\sigma(n\ln(R/\alpha) + \ln 10)}{\tau}. \tag{9}$$

Finally, we take $\tau$ such that the RHS of (9) equals to $\epsilon/2n$, which is equivalent to

$$\tau = \frac{2n\sigma(n\ln(R/\alpha) + \ln 10)}{\epsilon} = \tilde{O}(n^2/\epsilon).$$

This will finally promise that

$$\sup_{x \in \mathcal{K}} |F(x) - f(x)| \leq \max_{x \in \mathcal{N}_\alpha} |\tilde{\epsilon}_x| + \alpha L \leq \frac{\epsilon}{2n} + \frac{\epsilon}{2n} = \frac{\epsilon}{n}, \tag{10}$$

meeting the condition of Theorem 1. Consequently, with probability at least $0.9 \cdot 0.9 > 0.8$, we can find an $x^* \in \mathcal{K}$ such that $f(x^*) - \min_{x \in \mathcal{K}} f(x) \leq \epsilon$ using $\tilde{O}(n^3) \cdot \tau = \tilde{O}(n^5/\epsilon)$ queries to the quantum stochastic evaluation oracle (4). □

*Proof of Theorem 2.* We prove that Algorithm 2 satisfies Theorem 2.

Intuitively, we divide the $T$ rounds into $m + 1$ intervals where $m \leftarrow \lfloor \log_2 T \rfloor$, such that $[T] = \bigcup_{i=1}^{m+1} \mathcal{T}_i$ and $\mathcal{T}_i := \{2^{i-1}, 2^{i-1} + 1, \ldots, 2^i - 1\}$ for each $i \in [m]$. When executing in the interval $\mathcal{T}_i$, the output required by the online learner is always $x_t = x_{2^{i-1}}$, the $x$ at the end of the last interval. On the other hand, the queries in the current interval are applied to running the quantum stochastic convex optimization algorithm in Corollary 1 and output a nearly-optimal solution with probability at least $1 - O(1/T)$. With $|\mathcal{T}_i| = 2^{i-1}$ queries at hand, we divide them into $\log(TR)$ repeats of Corollary 1, each using $2^{i-1}/\log(TR)$ queries in the quantum algorithm. As a result, each repeat $j \in [\log(TR)]$ outputs a value $\tilde{x}_{2^i,j}$ such that

$$f(\tilde{x}_{2^i,j}) - \min_{x \in \mathcal{K}} f(x) \leq \tilde{O}(n^5 \log(TR)/2^{i-1})$$

with probability at least 0.8. We take $x_{2^i} := \arg\min_{j \in [\log T]} f(\tilde{x}_{2^i,j})$. With probability at least $1 - 0.8^{\log(TR)} = 1 - O(1/TR)$, we have

$$f(x_{2^i}) - \min_{x \in \mathcal{K}} f(x) \leq \tilde{O}(n^5 \log(TR)/2^i). \tag{11}$$

Going through all $i \in [m + 1]$ intervals, by the union bound, with probability at least

$$1 - (m + 1) \cdot O\left(\frac{1}{TR}\right) = 1 - O\left(\frac{\log T}{TR}\right),$$

---
**Algorithm 2** Quantum zeroth-order stochastic convex bandits
---
1: **procedure** QBANDITS($T$)
2:     $m \leftarrow \lfloor \log_2 T \rfloor, K \leftarrow \lfloor \log_2(TR) \rfloor$
3:     $x_1 \leftarrow 0$
4:     **for** $i \leftarrow 1, 2, \ldots, m+1$ **do**
5:         $t \leftarrow 2^{i-1}$
6:         **if** $t \leq m$ **then**
7:             $L \leftarrow 2^{i-1}$                            $\triangleright T_i := \{2^{i-1}, 2^{i-1}+1, \ldots, 2^i - 1\}$
8:         **else**
9:             $L \leftarrow T - 2^m + 1$                     $\triangleright T_{m+1} := \{2^m, 2^m+1, \ldots, T\}$
10:         **end if**
11:         **for** $j \leftarrow 1, 2, \ldots, K$ **do**
12:             $y_j \leftarrow$ QMINSTOCCONV($\mathcal{O}_f^{\tau,\alpha}, L/K$)         $\triangleright$ Corollary 1 with $|T_i|/K$ queries
13:             **for** $l \leftarrow 0, 1, \ldots, L/K$ **do**
14:                 $x_{t+l} \leftarrow x_{2^{i-1}}$         $\triangleright$ Onliner learner's output at time $t+l$
15:             **end for**
16:             **if** $f(y_j) < f(x_{2^i})$ **then**
17:                 $x_{2^i} \leftarrow y_j$
18:             **end if**
19:         **end for**
20:     **end for**
21: **end procedure**
---

we have

$$f(x_{2^i}) - \min_{x \in \mathcal{K}} f(x) \leq \tilde{O}(n^5 \log(TR)/2^{i-1}) \quad \forall i \in [m+1]. \tag{12}$$

In all, we get that the regret bound as desired:

$$
\begin{aligned}
\mathcal{R}_T &= \mathbb{E}\left[ \sum_{t=1}^{T} (f(x_t) - f^*) \right] \\
&\leq \left( 1 - O\left( \frac{\log T}{TR} \right) \right) \cdot \sum_{i=1}^{m+1} 2^{i-1} \cdot \tilde{O}\left( \frac{n^5 \log TR}{2^{i-1}} \right) + O\left( \frac{\log T}{TR} \right) \cdot T \cdot LR \\
&= \tilde{O}(n^5 \log(T) \log(TR)) + O(L \log T) \\
&= \tilde{O}(n^5 \log(T) \log(TR)). \qquad \square
\end{aligned}
$$

# Acknowledgement

We thank anonymous referees for valuable comments. TL was supported by a startup fund from Peking University. RZ was supported by the University Graduate Continuing Fellowship from UT Austin.

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
