**Roadmap.** In Appendix A, we provide more details of quantum walk and give our user-friendly framework. In Appendix B, we introduce the classical method for optimizing approximately convex functions in a self-contained way. In Appendix C, we prove our main result of quantum approximately convex optimization.

## A    Basic Facts about Quantum Walk

In this section, we first define the quantum walk operators and introduce some spectral properties. Then, we show how to efficiently implement a quantum walk.

### A.1    Definitions and spectral properties of quantum walk

Let $P$ be the transition operator of the classical Markov chain over the space $K$ such that

$$\int_K P(x,y)\mathrm{d}y = 1 \quad \forall x \in K.$$

We define the following states, which capture key properties of the quantum walk:

$$|\psi_x\rangle := \int_K \sqrt{P(x,y)} |y\rangle \, \mathrm{d}y \quad \forall x \in K.$$

**Definition 1** (Quantum walk operators). *The quantum walk uses the following three operators:*

- $U := \int_K |x\rangle |\psi_x\rangle \langle x| \langle 0| \, \mathrm{d}x$ *for any $x \in K$.*

- $\Pi := \int_K |x\rangle |\psi_x\rangle \langle x| \langle \psi_x| \, \mathrm{d}x$ *is the projection to the subspace* $\mathrm{span}\{|x\rangle |\psi_x\rangle\}_{x \in K}$.

- $S := \int_K \int_K |y\rangle |x\rangle \langle x| \langle y| \, \mathrm{d}x\mathrm{d}y$ *is to swap the two quantum registers.*

*Then, the quantum walk operator $W$ is defined by:*

$$W := S(2\Pi - I).$$

**Definition 2** (Alternative definition of quantum walk operator, [38]). *Define the quantum walk operator*

$$W' := U^\dagger S U R_A U^\dagger S U R_A,$$

*where $R_A$ denotes the reflection about the subspace $\mathcal{A} := \{|x\rangle |0\rangle \mid x \in K\}$ for random walk space $K$, $S$ is the swap operator, and $U$ is the following operator:*

$$U |x\rangle |0\rangle = \int_{y \in K} \sqrt{P(x,y)} |x\rangle |y\rangle \, \mathrm{d}y,$$

*for $P$ being the transition operator of the Markov chain.*

**Fact A.1** (Equivalence of the definitions, [14]). *Let $W$ be defined as in Definition 1 and let $W'$ be defined as in Definition 2. Then, $W'$ and $W$ have the same set of eigenvalues.*

**Fact A.2** ([14]). *Let $D$ be the discriminant operator of $P$ defined as $D(x,y) := \sqrt{P(x,y)P(y,x)}$. Then, $P$ and $D$ have the same set of eigenvalues.*

**Fact A.3** ([14]). *Let $\{\lambda_j\}$ be the eigenvalues of $D$. Then, the eigenvalues of $W$ are*

$$\left\{\pm 1, \lambda_j \pm \sqrt{1 - \lambda_j^2}i\right\}.$$

The following lemma shows that when the initial stationary is a warm start, then the eigenvalues whose eigenspaces have big overlap with the initial state are bounded away from 1.

**Lemma 5** (Effective spectral gap for warm start, [18, Lemma C.7]). *Let $M = (\Omega, p)$ be an ergodic reversible Markov chain with a transition operator $P$ and unique stationary state with a corresponding density $\rho$. Let $\{(\lambda_i, f_i)\}$ be the set of eigenvalues and eigenfunctions of $P$, and $|\psi_i\rangle$ be the eigenvectors of the corresponding quantum walk operator $W$. Let $\rho_0$ be a probability density that is*

*a warm start for $\rho$ and mixes up to TV-distance $\epsilon$ in $t$ steps of $M$. Furthermore, assume that $\rho_0$ is a $\beta$-warm start of $\rho$.*

*Let $|\phi_{\rho_0}\rangle$ be the resulting state of applying the quantum walk update operator $U$ to the state $|\rho_0\rangle$:*

$$|\phi_{\rho_0}\rangle = \int_\Omega \sqrt{\rho_0(x)} \int_\Omega \sqrt{P(x,y)} |x\rangle |y\rangle \,\mathrm{d}x\mathrm{d}y.$$

*Then, we have $|\langle\phi_{\rho_0}|\psi_i\rangle| = O(\beta\sqrt{\epsilon})$ for all $i$ with $1 > |\lambda_i| \geq 1 - O(1/t)$.*

Lemma 5 also applies to the initial distribution with a bounded $\ell_2$-warmness:

**Lemma 6** (Effective spectral gap for $\ell_2$-warm start, [13]). *Let $M = (\Omega, p)$ be an ergodic reversible Markov chain with a transition operator $P$ and unique stationary state with a corresponding density $\rho$. Let $\{(\lambda_i, f_i)\}$ be the set of eigenvalues and eigenfunctions of $P$, and $|\psi_i\rangle$ be the eigenvectors of the corresponding quantum walk operator $W$. Let $\rho_0$ be a probability density that is a warm start for $\rho$ and mixes up to TV-distance $\epsilon$ in $t$ steps of $M$. Furthermore, assume that $\|\rho/\rho_0\| = \int_\Omega \frac{\rho(x)}{\rho_0(x)}\rho(x)\mathrm{d}x \leq \gamma$ and $\|\rho_0/\rho\| = O(1)$.*

*Let $|\phi_{\rho_0}\rangle$ be the resulting state of applying the quantum walk update operator $U$ to the state $|\rho_0\rangle$:*

$$|\phi_{\rho_0}\rangle = \int_\Omega \sqrt{\rho_0(x)} \int_\Omega \sqrt{P(x,y)} |x\rangle |y\rangle \,\mathrm{d}x\mathrm{d}y.$$

*Then, we have $|\langle\phi_{\rho_0}|\psi_i\rangle| = O(\gamma^{1/4}\epsilon^{3/4} + \sqrt{\epsilon})$ for all $i$ with $1 > |\lambda_i| \geq 1 - O(1/t)$.*

## A.2 Efficient implementation of quantum walk

The goal of this section is to give a user-friendly quantum walk implementation cost-analysis (Theorem 5).

**Lemma 7** (Approximate reflector, [14, Corollary 4.1]). *Let $W$ be a unitary operator with a unique leading eigenvector $|\psi_0\rangle$ with eigenvalue 1. Denote the remaining eigenvectors by $|\psi_j\rangle$ with corresponding eigenvaluese $e^{2\pi i\xi_j}$ for $j \geq 1$. For any $\Delta \in (0,1]$ and $\epsilon < 1/2$, define $a := \log(1/\Delta)$ and $c := \log(1/\sqrt{\epsilon})$. Let $R$ be the reflector such that $R = \alpha|\psi_0\rangle\langle\psi_0| + (I - |\psi_0\rangle\langle\psi_0|)$.*

*For any constant $\alpha \in \mathbb{C}$, there exists a quantum circuit $\widetilde{R}$ that uses $a \cdot c$ ancilla qubits and invokes the controlled-$W$ gate $2^{a+1}c$ times such that*

- $\widetilde{R}|\psi_0\rangle|0\rangle^{\otimes ac} = R|\psi_0\rangle|0\rangle^{\otimes ac}$.

- $\left\|\widetilde{R}|\psi_j\rangle|0\rangle^{\otimes ac} - R|\psi_j\rangle|0\rangle^{\otimes ac}\right\|_2 \leq \sqrt{\epsilon}$ *for $j \geq 1$ with $\xi_j \geq \Delta$.*

**Lemma 8** ($\pi/3$-amplitude amplification, [38, Lemma 1]). *Let $|\psi\rangle, |\phi\rangle$ be two quantum states with $|\langle\psi|\phi\rangle| \geq p$ for some $p \in (0,1]$. Let $\omega = e^{i\pi/3}$. Define $R_\psi := \omega|\psi\rangle\langle\psi| + (I - |\psi\rangle\langle\psi|)$, and $R_\phi := \omega|\phi\rangle\langle\phi| + (I - |\phi\rangle\langle\phi|)$. Then, for $m \geq 1$, there exists a sequence of unitaries:*

$$V_0 = I, \;\; V_{j+1} = V_j R_\psi V_j^\dagger R_\phi V_j \;\; \forall j \in [m],$$

*such that*

$$|\langle\psi|V_m|\phi\rangle|^2 \geq 1 - (1-p)^{3^m}.$$

*Furthermore, the unitaries $R_\phi, R_\psi$ and their inverses are used at most $3^m$ times in $V_m$.*

**Theorem 5** (Quantum walk implementation cost). *Let $M_0, M_1$ be two ergodic reversible Markov chains with stationary distributions $\pi_0, \pi_1$, respectively. Suppose $\pi_0$ is $\beta_0$-warm with respect to $M_1$ and mixes up to total variation distance $\epsilon$ in $t_0(\epsilon)$ steps. Similarly, suppose $\pi_1$ is $\beta_1$-warm with respect to $M_0$ and mixes in $t_1(\epsilon)$ steps. Let $\beta := \max\{\beta_0, \beta_1\}$. Moreover, we assume that $|\langle\pi_0|\pi_1\rangle| \geq p$.*

*Given $|\pi_0\rangle$, we can obtain a state $|\widetilde{\pi}_1\rangle$ such that $\||\widetilde{\pi}_1\rangle - |\pi_1\rangle\|_2 \leq \epsilon$ using*

$$O\left(\sqrt{t_0(\epsilon p/\beta) + t_1(\epsilon p/\beta)} \cdot p^{-1} \cdot \log^2(1/(p\epsilon))\right)$$

*calls to the controlled walk operators controlled-$W_0'$, controlled-$W_1'$.*

*Proof.* By assumption, we know that $\pi_0$ mixes in $t'_0 = t_0(\epsilon_1/\beta)$ steps in $M_1$ to achieve total variation distance $\epsilon_1/\beta$, where $\epsilon_1$ is a parameter to be chosen later. Similarly, $\pi_1$ mixes in $t'_1 = t_1(\epsilon_1/\beta)$ steps in $M_0$ to achieve total variation distance $\epsilon_1/\beta$.

We start from $|\pi_0\rangle$. By Lemma 5, we have $|\pi_0\rangle = |\pi_{0,\mathsf{good}}\rangle + |e_0\rangle$, where $|\pi_{0,\mathsf{good}}\rangle$ lies in the subspace spanned by the eigenvectors $|\psi_j\rangle$ of $W'_1$ with corresponding eigenvalue $\lambda_j$ of $P_1$ such that $\lambda_j = 1$ or $\lambda_j \leq 1 - \Omega(1/t'_0)$. Let $e^{2\pi i \xi_j}$ be the eigenvalue of $|\psi_j\rangle$ of $W'_1$. By Fact A.1 and Fact A.3, we get that $\xi_j = 0$ or $\xi_j \geq \Omega(t_0'^{-1/2})$. By Lemma 5, we also have $\||e_0\rangle\| \leq \epsilon_1$.

Then, by Lemma 7 with $\Delta = \Omega(t_0'^{-1/2})$ and $\epsilon = \epsilon_1^2$, we can implement $\widetilde{R}_1$ such that $\|R_1 |\phi\rangle - \widetilde{R}_1 |\phi\rangle\|_2 \leq 2\epsilon_1$ using $O(\sqrt{t'_0}\log(1/\epsilon_1))$ calls to controlled-$W'_1$, where $|\phi\rangle$ is any state that occurs during $\pi/3$-amplitude amplification (Lemma 8) for $|\pi_0\rangle$ towards $|\pi_1\rangle$.

In the same way, we can start from $|\pi_1\rangle$ and show that $\widetilde{R}_0$ can be implemented using $O(\sqrt{t'_1}\log(1/\epsilon_1))$ calls to controlled-$W'_0$ such that $\|R_0 |\phi'\rangle - \widetilde{R}_0 |\phi'\rangle\| \leq 2\epsilon_1$, where $|\phi'\rangle$ is any state that occurs during $\pi/3$-amplitude amplification for $|\pi_1\rangle$ towards $|\pi_0\rangle$.

Suppose we can implement $R_0$ and $R_1$ perfectly. Then, we can prepare a state $|\widetilde{\pi}_1\rangle$ such that $|\langle\widetilde{\pi}_1|\pi_1\rangle| \geq 1 - (1-p)^{3^m}$ using $3^m$ calls to $R_0, R_1$ and their inverses, by applying $\pi/3$-amplitude amplification (Lemma 8) to $|\pi_i\rangle$. Thus, by taking $m = O(p^{-1}\log(1/\epsilon_2))$ where $\epsilon_2$ is a parameter to be chosen later, we have $\||\pi_1\rangle - |\widetilde{\pi}_1\rangle\|_2 \leq \epsilon_2$. However, since each call to $\widetilde{R}_0$ or $\widetilde{R}_1$ causes an error of $\epsilon_1$, the total error will be

$$O(\epsilon_2 + \epsilon_1 \cdot p^{-1}\log(1/\epsilon_2)) = \epsilon,$$

where we take $\epsilon_1 := O(p\epsilon\log^{-1}(1/\epsilon))$ and $\epsilon_2 := \epsilon_1^2$.

Therefore, the total number of calls to controlled-$W'_0$, controlled-$W'_1$ is

$$O\left((\sqrt{t'_0} + \sqrt{t'_1}) \cdot p^{-1}\log^2(1/p\epsilon)\right),$$

where $t'_i = t_i(\epsilon_1/\beta) = O(t_i(\epsilon p/\beta))$.

The theorem is then proved. □

The following corollary is an immediate consequence of Theorem 5, and it also gives Theorem 3.

**Corollary 2** (Quantum walk implementation cost ($\ell_2$-warm starts))**.** *Let $M_0, M_1$ be two ergodic reversible Markov chains with stationary distributions $\pi_0, \pi_1$, respectively. Suppose $\pi_0$ mixes towards $\pi_1$ in $M_1$ up to total variation distance $\epsilon$ in $t_0(\epsilon)$ steps. Similarly, suppose $\pi_1$ mixes towards $\pi_0$ in $M_0$ in $t_1(\epsilon)$ steps. Suppose $\|\pi_0/\pi_1\| = O(1)$ and $\|\pi_1/\pi_0\| = O(1)$. Moreover, we assume that $|\langle\pi_0|\pi_1\rangle| = \Omega(1)$.*

*Given $|\pi_0\rangle$, we can obtain a state $|\widetilde{\pi}_1\rangle$ such that $\||\widetilde{\pi}_1\rangle - |\pi_1\rangle\|_2 \leq \epsilon$ using*

$$O\left(\sqrt{t_0(\epsilon) + t_1(\epsilon)}\log^2(1/\epsilon)\right)$$

*calls to the controlled walk operators controlled-$W'_0$, controlled-$W'_1$.*

# B  Classical Approach for Optimizing Approximately Convex Functions

In this section, we introduce the classical approach [8] for the optimization of approximately convex functions as in Eq. (1).

## B.1  Low level: Hit-and-Run for approximate log-concave distributions

The Hit-and-Run walk uses a unidimensional rejection sampler to sample a point from the distribution $\pi_g$ restricted to a line $\ell$. The following lemma shows the performance guarantee of the unidimensional sampler:

**Lemma 9** (Unidimensional rejection sampler, [8, Lemma 5])**.** *Given $\beta = O(1)$. Let $g$ be a $\beta$-log-concave function and $\ell$ be a bounded line segment on $\mathcal{K}$. For $\epsilon \in (0, e^{-2\beta}/2)$, Algorithm 4 outputs a point $\mathbf{x} \in \ell$ with a distribution $\widetilde{\pi}_\ell$ such that*

$$d_{\mathrm{TV}}(\widetilde{\pi}_\ell, \pi_g|_\ell) \leq 3e^{2\beta}\epsilon.$$

---

**Algorithm 3** Hit-and-Run walk

---

1: **procedure** HITANDRUN($\pi_0, \pi_g, \Sigma, m$)       $\triangleright$ $\pi_g$ is the target distribution on $\mathcal{K}$ induced by a
    nonnegative function $g$, $\Sigma$ is a linear transformation
2:     $\mathbf{x}_0 \leftarrow$ sample from $\pi_0$
3:     Choose accuracy parameter $\epsilon_\ell$
4:     **for** $i \leftarrow 1, \ldots, m$ **do**
5:         $\mathbf{u} \leftarrow$ uniformly sample from the surface of ellipse given by $\Sigma$ acting on sphere
6:         $\ell(t) := \mathbf{x}_{i-1} + t\mathbf{u}$, compute $[\mathbf{s}, \mathbf{t}] \leftarrow \ell \cap \mathcal{K}$
7:         $\mathbf{x}_i \leftarrow$ UNISAMPLER($g, \beta, [\mathbf{s}, \mathbf{t}], \epsilon_\ell$)
8:     **end for**
9:     **return** $\mathbf{x}_m$
10: **end procedure**

---

*Moreover, the algorithm requires $\widetilde{O}(1)$ evaluations of the function $g$.*

The following theorem gives the mixing time of the standard Hit-and-Run walk for an approximate log-concave distribution, where we assume that in each step we directly sample from the restricted distribution $\pi_g|_\ell$.

**Theorem 6** (Mixing time of Hit-and-Run for approximate log-concave distribution, [8, Theorem 4])**.** *Let $\pi_g$ be the stationary measure associated with the Hit-and-Run walk based on a $\beta/2$-approximately log-concave function $g$, and let $\sigma^{(0)}$ be an initial distribution with $\ell_2$-warmness $M := \|\sigma^{(0)}/\pi_g\|$. There is a universal constant $C$ such that for any $\gamma \in (0, 1/2)$, if*

$$m \geq Cn^2 \frac{e^{6\beta}R^2}{r^2} \log^4 \left( \frac{e^\beta MnR}{r\gamma^2} \right) \log \left( \frac{M}{\gamma} \right),$$

*then $m$ steps of the Hit-and-Run random walk based on $g$ yield*

$$d_{\mathrm{TV}}(\sigma^{(m)}, \pi_g) \leq \gamma.$$

The next theorem shows the closeness between the output distribution of Algorithm 3 and the target distribution $\pi_g$. Due to the unidimensional rejection sampler (Algorithm 4), the stationary distribution of Algorithm 3 may not be exactly $\pi_g$. Nevertheless, we can still show that it will not deviate a lot.

**Theorem 7** (The effect of the rejection sampler, [8, Theorem 5])**.** *Let $\pi_g$, $\sigma^{(0)}$ be defined as in Theorem 6. Let $\hat{\sigma}^{(m)}$ denote the output distribution of Algorithm 3 with initial distribution $\hat{\sigma}^{(0)}$ in $m$ steps. Let $\epsilon_\ell$ be the accuracy parameter for the unidimensional rejection sampler (Algorithm 4). Then, we have*

$$d_{\mathrm{TV}}(\hat{\sigma}^{(m)}, \sigma^{(m)}) \leq m\epsilon_\ell + 2d_{\mathrm{TV}}(\hat{\sigma}^{(0)}, \sigma^{(0)}).$$

*In particular, for $\gamma \in (0, 1/e)$, suppose $d_{\mathrm{TV}}(\hat{\sigma}^{(0)}, \sigma^{(0)}) \leq \gamma/8$. Let $s \in (0, 1)$ be such that $H_s \leq \gamma/4$, where $H_s$ is defined to be:*

$$H_s := \sup_{A \subset \mathcal{K}: \pi_g(A) \leq s} |\pi_g(A) - \sigma^{(0)}(A)|.$$

*Then, there is a constant $C'$ such that, if we take $\epsilon_\ell := \gamma e^{-2\beta}/(12m)$ and*

$$m \geq C'n^2 \frac{e^{6\beta}R^2}{r^2} \log^4 \left( \frac{e^\beta nR}{rs} \right) \log(1/s),$$

*we have*

$$d_{\mathrm{TV}}(\hat{\sigma}^{(m)}, \pi_g) \leq \gamma.$$

### B.2 Mid level: rounding into isotropic position

The following lemma rounds a $\beta$-log-concave distribution to near-isotropic position.

**Algorithm 4** Unidimensional rejection sampler

1: **procedure** INITP($g$, $\beta$, $\ell = [\mathbf{s}, \mathbf{t}]$)
2:     **while** true **do**
3:         $\mathbf{x}_1 \leftarrow \frac{3}{4}\mathbf{s} + \frac{1}{4}\mathbf{t}$, $\mathbf{x}_2 \leftarrow \frac{1}{2}\mathbf{s} + \frac{1}{2}\mathbf{t}$, $\mathbf{x}_3 \leftarrow \frac{1}{4}\mathbf{s} + \frac{3}{4}\mathbf{t}$
4:         **if** $|\log(g(\mathbf{x}_1)) - \log(g(\mathbf{x}_3))| > \beta$ **then**
5:             $\mathbf{t} \leftarrow \mathbf{x}_3$ if $g(\mathbf{x}_1) > g(\mathbf{x}_3)$; $\mathbf{s} \leftarrow \mathbf{x}_1$ otherwise
6:         **else if** $|\log(g(\mathbf{x}_1)) - \log(g(\mathbf{x}_2))| > \beta$ **then**
7:             $\mathbf{t} \leftarrow \mathbf{x}_2$ if $g(\mathbf{x}_1) > g(\mathbf{x}_2)$; $\mathbf{s} \leftarrow \mathbf{x}_1$ otherwise
8:         **else if** $|\log(g(\mathbf{x}_2)) - \log(g(\mathbf{x}_3))| > \beta$ **then**
9:             $\mathbf{t} \leftarrow \mathbf{x}_3$ if $g(\mathbf{x}_2) > g(\mathbf{x}_3)$; $\mathbf{s} \leftarrow \mathbf{x}_2$ otherwise
10:         **else**
11:             **return** $\mathbf{p} \leftarrow \arg\max_{\mathbf{x} \in \{\mathbf{x}_1, \mathbf{x}_2, \mathbf{x}_3\}} g(\mathbf{x})$
12:         **end if**
13:     **end while**
14: **end procedure**
15: **procedure** BINSEARCH($g$, $\mathbf{x}_l$, $\mathbf{x}_r$, $V_l$, $V_r$)
16:     **while** true **do**
17:         $\mathbf{x}_m \leftarrow (\mathbf{x}_l + \mathbf{x}_r)/2$
18:         **if** $g(\mathbf{x}_m) > V_r$ **then**
19:             $\mathbf{x}_r \leftarrow \mathbf{x}_m$
20:         **else if** $g(\mathbf{x}_m) < V_l$ **then**
21:             $\mathbf{x}_l \leftarrow \mathbf{x}_m$
22:         **else**
23:             **return** $\mathbf{x}_m$
24:         **end if**
25:     **end while**
26: **end procedure**
27: **procedure** INITE($g$, $\beta$, $\ell = [\mathbf{s}, \mathbf{t}]$, $\mathbf{p}$, $\epsilon_\ell$)
28:     **if** $g(\mathbf{s}) \geq \frac{1}{2}e^{-\beta}\epsilon_\ell g(\mathbf{p})$ **then**
29:         $\mathbf{e}_0 \leftarrow \mathbf{s}$
30:     **else**
31:         $\mathbf{e}_0 \leftarrow$ BINSEARCH($g$, $\mathbf{s}$, $\mathbf{p}$, $\frac{1}{2}e^{-\beta}\epsilon_\ell g(\mathbf{p})$, $\epsilon_\ell g(\mathbf{p})$)
32:     **end if**
33:     **if** $g(\mathbf{t}) \geq \frac{1}{2}e^{-\beta}\epsilon_\ell g(\mathbf{p})$ **then**
34:         $\mathbf{e}_1 \leftarrow \mathbf{t}$
35:     **else**
36:         $\mathbf{e}_1 \leftarrow$ BINSEARCH($g$, $\mathbf{p}$, $\mathbf{t}$, $\frac{1}{2}e^{-\beta}\epsilon_\ell g(\mathbf{p})$, $\epsilon_\ell g(\mathbf{p})$)
37:     **end if**
38:     **return** $\mathbf{e}_0, \mathbf{e}_1$
39: **end procedure**
40: **procedure** UNISAMPLER($g$, $\beta$, $\ell = [\mathbf{s}, \mathbf{t}]$, $\epsilon_\ell$)
41:     $\mathbf{p} \leftarrow$ INITP($g$, $\beta$, $\ell$)
42:     $\mathbf{e}_0, \mathbf{e}_1 \leftarrow$ INITE($g$, $\beta$, $\ell$, $\mathbf{p}$, $\epsilon_\ell$)
43:     **while** true **do**
44:         $\mathbf{x} \leftarrow$ Uniform($[\mathbf{e}_0, \mathbf{e}_1]$), $r \leftarrow$ Uniform($[0, 1]$)
45:         **if** $r \leq g(\mathbf{x})/(e^{3\beta}g(\mathbf{p}))$ **then**
46:             **return** $\mathbf{x}$
47:         **end if**
48:     **end while**
49: **end procedure**

**Lemma 10** (Rounding $\beta$-log-concave distribution, [8, Lemma 9]). *Let $\pi$ be a $\beta$-log-concave distribution in $\mathbb{R}^n$. By taking $N = \Theta(n \log n)$ i.i.d. samples $\mathbf{x}_1, \ldots, \mathbf{x}_n$ from $\pi$, we have*

$$\frac{1}{2} \le \sigma_{\min}\left(\frac{1}{N} \sum_{i \in [N]} \mathbf{x}_i \mathbf{x}_i^\top\right) \le \sigma_{\max}\left(\frac{1}{N} \sum_{i \in [N]} \mathbf{x}_i \mathbf{x}_i^\top\right) \le \frac{3}{2}$$

*holds with probability at least $1 - n^{-O(1)}$.*

### B.3  High level: simulated annealing

At high level, we run a simulated annealing for a series of functions:

$$h_i(x) := \exp(-f(x)/T_i), \quad \text{and} \quad g_i(x) := \exp(-F(x)/T_i),$$

where $f, F$ satisfy Eq. (1) and $\{T_i\}_{i \in [K]}$ are parameters to be chosen later.

---

**Algorithm 5** Simulated annealing

---

1: **procedure** SIMANNEALING($K, \{T_i\}_{i \in [K]}$)
2:     $N \leftarrow \Theta(n \log n)$                                    ▷ The number of strands
3:     $X_0^j \sim \text{Uniform}(\mathcal{K})$ for $j = 1, \ldots, N$
4:     $\mathcal{K}_0 \leftarrow \mathcal{K}, \Sigma_0 \leftarrow I$
5:     $m \leftarrow \widetilde{O}(n^3)$                                    ▷ Theorem 7
6:     **for** $i \leftarrow 1, \ldots, K$ **do**
7:         $\Sigma_i' \leftarrow$ the rounding linear transformation for $\{X_{i-1}^j\}_{j \in [N]}$
8:         $\Sigma_i \leftarrow \Sigma_i' \circ \Sigma_{i-1}$
9:         **for** $j \leftarrow 1, \ldots, N$ **do**
10:             $X_i^j \leftarrow \text{HITANDRUN}(X_{i-1}^j, \pi_{g_i}, \Sigma_i, m)$            ▷ Algorithm 3
11:         **end for**
12:     **end for**
13:     **return** $\arg\min_{i \in [K], j \in [N]} F(X_i^j)$
14: **end procedure**

---

**Lemma 11** (The warmness of annealing distributions, [8, Lemma 8]). *Let $g(x) = \exp(-F(x))$ be a $\beta$-log-concave function. Let $\pi_{g_i}$ be a distribution with density proportional to $g_i(x) = \exp(-F(x)/T_i)$, supported on $\mathcal{K}$. Let $T_i := T_{i-1}\left(1 - \frac{1}{\sqrt{n}}\right)$. Then,*

$$\|\pi_{g_i}/\pi_{g_{i+1}}\| \le C_\gamma = 5\exp(2\beta/T_i).$$

**Theorem 8** (Sample Guarantee for the simulated annealing, [8, Theorem 6]). *Fix a target accuracy $\gamma \in (0, 1/e)$ and let $g$ be an $\beta/2$-approximately log-concave function in $\mathbb{R}^n$. Suppose the simulated annealing algorithm (Algorithm 5) is run for $K = \sqrt{n} \log(1/\rho)$ epochs with temperature parameters $T_i = (1 - 1/\sqrt{n})^i$ for $0 \le i \le K$. If the Hit-and-Run with the unidimensional sampling scheme (Algorithm 3) is run for $m = \widetilde{O}(n^3)$ number of steps prescribed in Theorem 7, the algorithm maintains that*

$$d_{\mathrm{TV}}(\hat{\sigma}_i^{(m)}, \pi_{g_i}) \le e\gamma$$

*for each $i \in [K]$, where $\hat{\sigma}_i^{(m)}$ is the distribution of the $m$-th step of Hit-and-Run. Here, $m$ depends polylogarithmically on $1/\rho$.*

Then, we have the following optimization guarantee for the simulated annealing procedure:

**Theorem 9** (Optimization guarantee for the simulated annealing, [8, Corollary 1]). *Suppose $F$ is approximately convex and $|F - f| \le \epsilon/n$ as in Eq. (1). The simulated annealing method with $K = \sqrt{n} \log(n/\epsilon)$ epochs produces a random point $X$ such that*

$$\mathbb{E}[f(X)] - \min_{\mathbf{x} \in \mathcal{K}} f(\mathbf{x}) \le \epsilon,$$

*and thus,*

$$\mathbb{E}[F(X)] - \min_{\mathbf{x} \in \mathcal{K}} F(\mathbf{x}) \le 2\epsilon.$$

*Furthermore, the number of oracle queries required by the method is $\widetilde{O}(n^{4.5})$.*

# C  Quantum Speedup for Optimizing Approximately Convex Functions

As we discussed in previous section, there are three levels for the optimization algorithm. The goal of this section is to prove Theorem 1, where we improve the classical query complexity $\widetilde{O}(n^{4.5})$ (Theorem 9) to quantum query complexity $\widetilde{O}(n^3)$. The main idea is to use quantum walk algorithm (introduced in Appendix A) to speed-up the low level such that each sample can be generated with less queries.

## C.1  Quantum speedup for low-level

In this section, we show how to use the quantum walk algorithm to speedup the sampling procedure in the simulated annealing process. According to the framework (Corollary 2), we first show that the each Markov chain's stationary distribution in the annealing process is a warm-start for its adjacent chains, and the Markov chains are slowly-varying. Then, we show how to implement the quantum walk operator for the Hit-and-Run walk. Finally, we prove the quantum speedup from $\widetilde{O}(n^3)$ classical query complexity to $\widetilde{O}(n^{1.5})$ quantum query complexity.

**Warmness and overlap for the stationary distributions.**  We first show that $\pi_{g_i}$ is a warm-start for $\pi_{g_{i+1}}$, and vice versa.

By lemma 11, we know that $\|\pi_{g_i}/\pi_{g_{i+1}}\| \leq 5\exp(2\beta/T_i)$. Similarly, we can also bound $\|\pi_{g_{i+1}}/\pi_{g_i}\|$:

**Lemma 12.** *Let $g(x) = exp(-F(x))$ be a $\beta$-log-concave function. Let $\pi_{g_i}$ be a distribution with density proportional to $g_i(x) = \exp(-F(x)/T_i)$, supported on $\mathcal{K}$. Let $T_i := T_{i-1}\left(1 - \frac{1}{\sqrt{n}}\right)$. Then,*

$$\|\pi_{g_{i+1}}/\pi_{g_i}\| \leq 8\exp(2\beta/T_{i+1}).$$

*Proof.* Define $Y(a) := \int_{\mathcal{K}} \exp(-F(x)a)\mathrm{d}x$. Then, we have

$$\|\pi_{g_{i+1}}/\pi_{g_i}\| = \frac{\int_{\mathcal{K}} \exp(-F(x)(2/T_{i+1} - 1/T_i))\mathrm{d}x \cdot \int_{\mathcal{K}} \exp(-F(x)/T_i)\mathrm{d}x}{\left(\int_{\mathcal{K}} \exp(-F(x)/T_{i+1})\mathrm{d}x\right)^2}$$

$$= \frac{Y(2/T_{i+1} - 1/T_i)Y(1/T_i)}{Y(1/T_{i+1})^2}.$$

Define $G(x,t) := g(x/t)^t$. Then, we have

$$G(\lambda x + (1-\lambda)x', \lambda t + (1-\lambda)t') = g\left(\frac{\lambda x + (1-\lambda)x'}{\lambda t + (1-\lambda)t'}\right)^{\lambda t + (1-\lambda)t'}$$

$$= g\left(\frac{\lambda t}{\lambda t + (1-\lambda)t'}\frac{x}{t} + \frac{(1-\lambda)t'}{\lambda t + (1-\lambda)t'}\frac{x'}{t'}\right)^{\lambda t + (1-\lambda)t'}$$

$$\geq \exp(-\beta(\lambda t + (1-\lambda)t')) \cdot g\left(\frac{x}{t}\right)^{\lambda t} \cdot g\left(\frac{x'}{t'}\right)^{(1-\lambda)t'}$$

$$= \exp(-\beta(\lambda t + (1-\lambda)t')) \cdot G(x,t)^{\lambda} \cdot G(x',t')^{1-\lambda}$$

$$= (\exp(-\beta t)G(x,t))^{\lambda} \cdot (\exp(-\beta t')G(x',t'))^{1-\lambda},$$

where the inequality follows from $g$ is $\beta$-log-concave.

By Prékopa–Leindler inequality (Theorem 10), it implies that

$$\int_{\mathcal{K}} G(x, \lambda t + (1-\lambda)t')\mathrm{d}x \geq \left(\int_{\mathcal{K}} \exp(-\beta t)G(x,t)\mathrm{d}x\right)^{\lambda} \cdot \left(\int_{\mathcal{K}} \exp(-\beta t')G(x,t')\mathrm{d}x\right)^{1-\lambda}.$$

Note that

$$\int_{\mathcal{K}} G(x,t)\mathrm{d}x = \int_{\mathcal{K}} g\left(\frac{x}{t}\right)^t \mathrm{d}x = t^n \int_{\mathcal{K}} g(x)^t \mathrm{d}x = t^n \int_{\mathcal{K}} \exp(-F(x)t)\mathrm{d}x = t^n Y(t).$$

Hence, for $\lambda = \frac{1}{2}$, we have

$$\left(\frac{t+t'}{2}\right)^{2n} Y\left(\frac{t+t'}{2}\right)^2 \geq \exp(-\beta(t+t')/2) \cdot t^n Y(t) \cdot t'^n Y(t'),$$

which implies that

$$\frac{Y(t)Y(t')}{Y(\frac{t+t'}{2})^2} \leq \exp\left(\frac{\beta(t+t')}{2}\right) \cdot \left(\frac{(t+t')^2/4}{tt'}\right)^n. \tag{13}$$

By taking $t = 2/T_{i+1} - 1/T_i$ and $t' = 1/T_i$, we have

$$\|\pi_{g_{i+1}}/\pi_{g_i}\| \leq \frac{Y(2/T_{i+1} - 1/T_i)Y(1/T_i)}{Y(1/T_{i+1})^2}$$

$$\leq \exp(2\beta/T_{i+1}) \cdot \left(\frac{(1/T_{i+1})^2}{(2/T_{i+1} - 1/T_i)(1/T_i)}\right)^n$$

$$= \exp(2\beta/T_{i+1}) \cdot \left(\frac{1}{(2 - (1 - 1/\sqrt{n}))(1 - 1/\sqrt{n})}\right)^n$$

$$= \exp(2\beta/T_{i+1}) \cdot \left(1 + \frac{1}{n-1}\right)^n$$

$$\leq \exp(2\beta/T_{i+1}) \cdot \exp(n/(n-1))$$

$$\leq 8\exp(2\beta/T_{i+1}),$$

where the third step follows from $T_{i+1} = T_i(1 - \frac{1}{\sqrt{n}})$.

The lemma is then proved. $\qquad\qquad\square$

**Remark 1.** *Since we assume that $|F(x) - f(x)| \leq \epsilon/n$ in Eq. (1), i.e., $\beta = \epsilon/n$, by Lemmas 11 and 12, we know that the warmness $M := \max\{\|\pi_{g+i}/\pi_{g_{i+1}}\|, \|\pi_{g_{i+1}}/\pi_{g_i}\|\}$ can be bounded by $O(\exp(2\epsilon/(nT_{i+1})))$. Since we choose the final temperature $T_k = \epsilon/n$, we get that $M = O(1)$. Therefore, it satisfies the warmness condition in Corollary 2.*

**Theorem 10** (Prékopa–Leindler inequality, [32, 33]). *Let $0 < \lambda < 1$ and let $f, g, h : \mathbb{R}^n \to [0, \infty)$ be measurable functions. Suppose that these functions satisfy*

$$h(\lambda x + (1 - \lambda)y) \geq f(x)^\lambda \cdot g(y)^{1-\lambda} \quad \forall x, y \in \mathbb{R}^n.$$

*Then, we have*

$$\int_{\mathbb{R}^n} h(x)\mathrm{d}x \geq \left(\int_{\mathbb{R}^n} f(x)\mathrm{d}x\right)^\lambda \cdot \left(\int_{\mathbb{R}^n} g(x)\mathrm{d}x\right)^{1-\lambda}.$$

**Lemma 13** (Bound distribution overlap). *Let $g(x) = \exp(-F(x))$ be a $\beta$-log-concave function. Let $\pi_{g_i}$ be a distribution with density proportional to $g_i(x) = \exp(-F(x)/T_i)$, supported on $\mathcal{K}$. Let $T_i := T_{i-1}\left(1 - \frac{1}{\sqrt{n}}\right)$. Then,*

$$\langle\pi_i|\pi_{i+1}\rangle \geq \exp(-(\beta/T_{i+1} + 1)/2).$$

*Proof.* We can write the overlap as follows:

$$\langle\pi_{g_i}|\pi_{g_{i+1}}\rangle = \frac{\int_{\mathcal{K}} \sqrt{g_i(x)g_{i+1}(x)}\mathrm{d}x}{(\int_{\mathcal{K}} g_i(x)\mathrm{d}x)^{1/2} \cdot (\int_{\mathcal{K}} g_{i+1}(x)\mathrm{d}x)^{1/2}}$$

$$= \frac{\int_{\mathcal{K}} \exp(-F(x)(1/T_i + 1/T_{i+1})/2)\mathrm{d}x}{(\int_{\mathcal{K}} \exp(-F(x)/T_i)\mathrm{d}x)^{1/2} \cdot (\int_{\mathcal{K}} \exp(-F(x)/T_{i+1})\mathrm{d}x)^{1/2}}$$

$$= \frac{Y((1/T_i + 1/T_{i+1})/2)}{Y(1/T_i)^{1/2}Y(1/T_{i+1})^{1/2}},$$

where $Y(t) := \int_{\mathcal{K}} \exp(-F(x)t)\mathrm{d}x$.

By Eq. (13), we have

$$\frac{Y(1/T_i)Y(1/T_{T_{i+1}})}{Y((1/T_i+1/T_{i+1})/2)^2} \le \exp(\beta(1/T_i+1/T_{i+1})/2) \cdot \Big(\frac{(1/T_i+1/T_{i+1})^2/4}{1/(T_iT_{i+1})}\Big)^n$$

$$= \exp\big(\beta(2-1/\sqrt{n})/(2T_{i+1})\big) \cdot \Big(1+\frac{1}{4(n-\sqrt{n})}\Big)^n$$

$$\le \exp\Big(\frac{1}{4}\frac{\sqrt{n}}{\sqrt{n}-1}\Big) \cdot \exp(\beta/T_{i+1})$$

$$\le \exp(\beta/T_{i+1}+1),$$

where the second step follows from $T_{i+1} = T_i(1-1/\sqrt{n})$.

Therefore,

$$\langle \pi_{g_i}|\pi_{g_{i+1}}\rangle \ge \exp(-(\beta/T_{i+1}+1)/2).$$

$\square$

**Remark 2.** *By taking $\beta = \epsilon/n$ and $T_i \ge \epsilon/n$ in Lemma 13, we have for any $i \in [K-1]$, the overlap can be upper-bounded by:*

$$\langle \pi_{g_i}|\pi_{g_{i+1}}\rangle \ge \exp(-(\beta/T_K+1)/2) = e^{-1}.$$

**Implementing the quantum walk operator.** We introduce how to implement the quantum walk update operator $U$ such that:

$$U|x\rangle|0\rangle = \int_{\mathcal{K}} \sqrt{P_{x,y}}\,|x\rangle|y\rangle\,\mathrm{d}y,$$

where $P$ is the stochastic transition matrix for the Hit-and-Run walk.

Given an input state $|x\rangle$. We first prepare an $n$-dimensional Gaussian state in an ancilla register:

$$|x\rangle|0\rangle \longrightarrow |x\rangle \int_{\mathbb{R}^n} (2\pi)^{-n/4}|z\rangle\,\mathrm{d}z.$$

Then, by normalizing $z$ and applying the linear transformation $\Sigma$ in another quantum register, we get that

$$|x\rangle \int_{\mathbb{R}^n} (2\pi)^{-n/4}|z\rangle\Big|\frac{\Sigma z}{\|z\|}\Big\rangle\,\mathrm{d}z.$$

If we un-compute the $|z\rangle$ register, we get that (ignoring the normalization factor):

$$|x\rangle \int_{\Sigma\mathbb{S}^n} |u\rangle\,\mathrm{d}u.$$

Next, we coherently compute the two end-points of $\ell \cap \mathcal{K}$ for $\ell(t) := x+ut$ in the ancilla registers:

$$\int_{\Sigma\mathbb{S}^n} \mathrm{d}u\,|x\rangle|u\rangle|0\rangle \longrightarrow \int_{\Sigma\mathbb{S}^n} \mathrm{d}u\,|x\rangle|u\rangle|s,t\rangle$$

We coherently simulate the unidimensional sampler (Algorithm 4). More specifically, we can compute the points $p, e_0, e_1$ in ancilla registers:

$$\int_{\Sigma\mathbb{S}^n} \mathrm{d}u\,|x\rangle|u\rangle|s,t,p,e_0,e_1\rangle$$

Then, we prepare two unifrom distribution states in the next two ancilla qubits:

$$\int_{\Sigma\mathbb{S}^n} \mathrm{d}u\,|x\rangle|u\rangle|s,t,p,e_0,e_1\rangle \int_{[0,1]^2} |r',r\rangle\,\mathrm{d}r'\mathrm{d}r$$

And the next proposed point $y$ can be computed via $y := e_0 + r'(e_1-e_0)$:

$$\int_{\Sigma\mathbb{S}^n} \mathrm{d}u\,|x\rangle|u\rangle|s,t,p,e_0,e_1\rangle \int_{[0,1]^2} |r',r\rangle\,\mathrm{d}r'\mathrm{d}r\,|y\rangle$$

Then, we check the condition $r \leq g(y)/(3^\beta g(p))$ by querying the evaluation oracle twice and use an ancilla qubit to indicate whether it is satisfied:

$$\int_{\Sigma \mathbb{S}^n} \int_{[0,1]^2} \mathrm{d}u \mathrm{d}r' \mathrm{d}r \, |x\rangle \, |u\rangle \, |s,t,p,e_0,e_1\rangle \, |r',r\rangle \, |y\rangle \, |b\rangle ,$$

where $b \in \{0,1\}$. Then, we post-select[4] the last qubit for $b = 1$. By un-computing the registers for $u, s, t, p, e_0, e_1, r, r'$, we get the desired state:

$$|x\rangle \int_{\mathcal{K}} \sqrt{P_{x,y}} \, |y\rangle .$$

By Lemma 9, we get that this procedure (including the post-selection cost) takes $O(1)$ oracle queries with high probability. Therefore, we get the following lemma:

**Lemma 14** (Implementation cost of the quantum walk update operator). *For the Hit-and-Run walk (Algorithm 3) with the unidimensional sampler (Algorithm 4), the quantum walk update operator $U$ can be implemented by querying the evaluation oracle $O(1)$ times.*

$\widetilde{O}(n^{1.5})$**-query quantum algorithm.** We have the following theorem:

**Theorem 11** (Quantum speedup for the Hit-and-Run sampler). *Let $\gamma \in (0, 1/e)$. Let $\pi_g$ be the stationary measure associated with the Hit-and-Run walk based on a $\beta/2$-approximately log-concave function g. Let $T_i = (1 - 1/\sqrt{n})^i$ for $0 \leq i \leq K$ be the annealing schedule. Suppose we use the quantum walk to implement the Hit-and-Run walk (Algorithm 3). Then, for each $0 \leq i \leq K - 1$, given a state $|\pi_{g_i}\rangle$, we can produce a state $|\hat{\sigma}_i^{(m)}\rangle$ such that*

$$\||\pi_{g_{i+1}}\rangle - |\hat{\sigma}_i^{(m)}\rangle\|_2 \leq O(\gamma),$$

*using $m = \widetilde{O}(n^{1.5})$ calls for the evaluation oracle.*

*Proof.* We use the quantum walk framework in Corollary 2.

For the warmness, by Remark 1, we know that in this annealing schedule, $\|\pi_{g_i}/\pi_{g_{i+1}}\|$ and $\|\pi_{g_{i+1}}/\pi_{g_i}\|$ are upper-bounded by some constants.

Then, by Theorem 6, we get that the number of steps for evolving from $\pi_{g_i}$ to $\pi_{g_{i+1}}$ and from $\pi_{g_{i+1}}$ to $\pi_{g_i}$ is $\widetilde{O}(n^3)$ classically. The proof of Theorem 7 implies that the stationary distribution of the Hit-and-Run walk with unidimensional sampler is very close to the original Markov chain, only causing a constant blowup to the total variation distance. Thus, for $\gamma' = O(\gamma)$, we have $t_1(\gamma'), t_2(\gamma') = \widetilde{O}(n^3)$ in Corollary 2.

By Remark 2, we know that in this annealing schedule, the adjacent distributions have a big overlap. In particular, we have $|\langle \pi_{g_i} | \pi_{g_{i+1}} \rangle| \geq \Omega(1)$, satisfying the condition in Corollary 2.

Therefore, by Corollary 2, we get that the state $|\hat{\sigma}_i^{(m)}\rangle$ satisfying $\||\pi_{g_{i+1}}\rangle - |\hat{\sigma}_i^{(m)}\rangle\|_2 \leq O(\gamma)$ can be prepared using $\widetilde{O}(n^{1.5})$ calls to the controlled walk operators.

By Lemma 14, each call to the quantum walk operator can be implemented with $O(1)$ query to the evaluation oracle. Hence, the total query complexity is $\widetilde{O}(n^{1.5})$.

The theorem is then proved. $\square$

## C.2   Non-destructive rounding in the mid-level

In the middle level, we need to compute the linear transformation $\Sigma_i$ that rounds the $\beta$-logconcave distribution to near-isotropic position. Moreover, we are given access to $N$ copies of the quantum states $|\pi_{g_i}\rangle$ and we will compute $\Sigma_i$ in a non-destructive way.

---

[4]We can measure the last qubit. If the measurement outcome is 0, we reinitialize the $r, r'$ registers and re-preapre $|y\rangle$ and $|b\rangle$. We repeat this process until we measure $b = 1$.

Classically, by Lemma 10, we can take

$$\Sigma_i'(x^1, \ldots, x^N) := \frac{1}{N} \sum_{i=1}^{N} x^j i^{j\top},$$

where $x_i^j$ is the $j$-th independent sample from $\pi_{g_i}$. Then, the linear transformation in the $i$-th iteration is $\Sigma_i$ composite with the linear transformation in the $(i-1)$-th iteration, i.e.,

$$\Sigma_i(x_1, \ldots, x_n) := \Sigma_i'(x_1, \ldots, x_N) \cdot \Sigma_{i-1}.$$

In quantum, we can use a quantum circuit to simulate the classically computation for $\Sigma_i'$ and $\Sigma_i$ coherently, which computes the following superposition state:

$$\int_{\mathcal{K}} dx^1 \cdots \int_{\mathcal{K}} dx^N \prod_{j=1}^{N} \sqrt{\pi_{g_i}(x^j)} \cdot |x^1\rangle \cdots |x^N\rangle |\Sigma_i(x_1, \ldots, x^n)\rangle. \tag{14}$$

That is, the first $N$ quantum registers contain $N$ copies of the state $|\pi_{g_i}\rangle$, and the last quantum register contains the linear transformation $\Sigma_i$. If we directly measure the last register, we can get the desired matrix, but the coherence of the quantum states $|\pi_{g_i}\rangle$ are also destroyed.

To resolve this issue, we use the following theorem of Harrow and Wei:

**Theorem 12** (Non-destructive amplitude estimation, [21]). *Let $P$ be an observable. Given state $|\psi\rangle$ and reflections $R_\psi = 2|\psi\rangle\langle\psi| - I$ and $R = 2P - I$, and any $\eta > 0$, there exists a quantum algorithm that outputs $\widetilde{a}$, an approximation to $a := \langle\psi|P|\psi\rangle$, so that*

$$|a - \widetilde{a}| \leq 2\pi \frac{a(1-a)}{M} + \frac{\pi^2}{M^2}.$$

*with probability at least $1 - \eta$ and $O(\log(1/\eta)M)$ uses of $R_\psi$ and $R$. Morover the algorithm restores the state $|\psi\rangle$ with probability at least $1 - \eta$.*

Then, we can create $O(\log N)$ copies of the state in Eq. (14), and non-destructively estimate the mean of the last quantum register via the procedure in [14, 21]. More specifically, we start from $\widetilde{O}(N)$ copies of the states $|\pi_{g_{i-1}}\rangle$, and evolve them to $|\pi_{g_i}\rangle$. In the same time, the reflection operator $R$ can be approximately implemented by Lemma 7. Then, the mean value can be estimated by Theorem 12. Note that we can estimate all the coordinates of $\Sigma_i$ in the same time using the non-destructive mean estimation quantum circuit. And we get that the success probability of this procedure is at least $1 - 1/\text{poly}(N)$. After that, the states in the first $N$ registers will be restored. Therefore, we get that:

**Lemma 15** (Non-destructive rounding). *For $i \in [K]$, the linear transformation $\Sigma_i$ at the $i$-iteration of the annealing process (Algorithm 5) can be obtained using $\widetilde{O}(N)$ copies of the states $|\pi_{g_{i-1}}\rangle$, with query complexity $\widetilde{O}(N \cdot \mathcal{C})$ where $\mathcal{C}$ is the cost of evolving $|\pi_{g_{i-1}}\rangle$ to $|\pi_{g_i}\rangle$. Moreover, the states $|\pi_{g_{i-1}}\rangle$ will be recovered with high probability.*

### C.3 Proof of Theorem 1

*Proof of Theorem 1.* The quantum algorithm for optimizing an approximately convex function is given in Algorithm 6. By Theorem 11 and Lemma 15, we know that it has the same optimization guarantee as the classical procedure (Algorithm 5). Thus, we take $K = \sqrt{n}\log(n/\epsilon)$. And the output $x_*$ of QSIMANNEALING procedure satisfies:

$$F(x_*) - \min_{x \in \mathcal{K}} F(x) \leq O(\epsilon)$$

with high probability.

Then, consider the query complexity. We have $K = \sqrt{n}\log(n/\epsilon)$ stages in the annealing process. In each iteration, the quantum walk has query complexity $\mathcal{C} = \widetilde{O}(n^{1.5})$ by Theorem 11. Thus, the query cost of Line 5 is $\widetilde{O}(N\mathcal{C}) = \widetilde{O}(n^{2.5})$. Also, the query cost of Line 6 is also $\widetilde{O}(n^{2.5})$. Therefore, the total query complexity of the annealing procedure is

$$K \cdot \widetilde{O}(n^{2.5}) = \widetilde{O}(n^3).$$

$\square$

---

**Algorithm 6** Quantum speedup for approximately convex optimization.

---

1: **procedure** QSimAnnealing($K, \{T_i\}_{i \in [K]}$)
2:      $N \leftarrow \widetilde{O}(n)$                                                         ▷ The number of strands
3:      Prepare $N$ (approximately) copies of $|\pi_0\rangle$, denoted as $|\widetilde{\pi}_0^{(1)}\rangle, \ldots, |\widetilde{\pi}_0^{(N)}\rangle$, where $\pi_0 =$ Uniform($\mathcal{K}$)
4:      **for** $i \leftarrow 1, \ldots, K$ **do**
5:          Use the $N$ copies of the state $|\pi_{i-1}\rangle$ to nondestructively obtain the linear transformation $\Sigma_i$. Let $|\hat{\pi}_{i-1}^{(1)}\rangle, \ldots, |\hat{\pi}_{i-1}^{(N)}\rangle$ denote the post-measurements states            ▷ Lemma 15
6:          Apply quantum walk with $\Sigma_i$ to evolve the states $|\hat{\pi}_{i-1}^{(1)}\rangle, \ldots, |\hat{\pi}_{i-1}^{(N)}\rangle$ to $|\widetilde{\pi}_i^{(1)}\rangle, \ldots, |\widetilde{\pi}_i^{(N)}\rangle$                                              ▷ Theorem 11
7:      **end for**
8:      $x_K^j \leftarrow$ measure the final state $|\widetilde{\pi}_K^{(j)}\rangle$ for $j \in [N]$
9:      **return** $\arg\min_{j \in [N]} F(x_K^j)$
10: **end procedure**

---