# OpenReview forum: "Quantum Speedups of Optimizing Approximately Convex Functions with Applications to Logarithmic Regret Stochastic Convex Bandits"
_NeurIPS.cc/2022/Conference — NeurIPS 2022 Accept_

### Official Review · Reviewer_bFZB · 2022-07-11

**Rating:** 7
**Confidence:** 1
**Soundness:** 2 fair
**Presentation:** 2 fair
**Contribution:** 2 fair

**Summary:**

The paper studies the quantum algorithm for minimizing an approximately convex function. It is faster than the classical algorithm by a factor  O(n^1.5). This is achieved by implementing a quantum version of the classical algorithm which is based on simulated annealing and the hit-and-run walk. It also gives rise to a quantum algorithm for zero-th order stochastic convex bandits with an exponential speedup in time horizon.

**Questions:**

This paper focuses on zero-th order quantum optimization for the considered problem. I am curious about if there exists any result on first-order quantum optimization.

**Limitations:**

I feel the presentation could be improved a lot so that it is more accessible to general readers. It would better if Eq. (4) can be explained more: how to understand the right-hand side which is a quantum state followed by another quantum state (there seem to be no such definitions in Section 2.1). Also,  $\xi$ is n-dimensional but $y$ is a scalar.


**Strengths And Weaknesses:**

originality: the paper initiates the quantum version of minimizing approximately convex functions.

quality: technical depth is sufficient

clarity: not friendly to readers not in this area.

significance: pure theoretical contribution

---

> ### Author Response · Authors · 2022-07-29
> **Official Reply**
>
> We thank the reviewer for the general positive opinions about our paper. The question about results on first-order quantum optimization is a very natural one, considering that many existing classical optimization algorithms are first-order methods. However, there are two important reasons why in this paper we stick to zeroth-order quantum algorithms for optimizing approximately convex functions.
>
> - On the one hand, when we talk about an approximately convex function F following the definition in Eq. (1), although its eps/n-close to a convex function f, F per se might not even be continuous, so it does not make much sense to talk about first-order and high-order optimization methods for approximate convex optimization, and we have to use the noisy evaluation (i.e. zeroth-order) oracle. This is also the circumstance in seminal papers for optimizing approximately convex functions, including Belloni et al. (Ref. [8]) and Risteski and Li (Ref. [32]).
>
> - On the other hand, in quantum optimization algorithms, a powerful tool is Jordan’s quantum algorithm for gradient computation (https://journals.aps.org/prl/abstract/10.1103/PhysRevLett.95.050501), which uses quantum Fourier transform to compute the gradient of a function at a specific point only using O(1) quantum evaluation queries to the function. Intuitively, this means that zeroth-order quantum optimization algorithms are at least as powerful as first-order classical optimization algorithms, and having this in hand, most existing quantum optimization algorithms adopt the zeroth-order quantum oracle, including Refs. [5,14,15,18,39].
>
> Nevertheless, in future versions, we will clarify the comparison between zeroth-order and first-order optimization algorithms following the discussions above.
>
> In future versions, we will also try our best to increase the clarity of our paper and make it more accessible to general readers by adding explanations in favor of readers not working in quantum computing. Regarding Eq. (4), the right-hand side can be understood as the tensor product of the two quantum states (the $\otimes$ operator is typically omitted for being more concise). This follows the same notation as in Eq. (3), and the discussion between Eq. (3) and (4): we can take input states of form $\sum_{i=1}^{m} c_{i} |x_{i}\rangle \otimes |0\rangle$ where $x_{i}\in\mathbb{R}^{n}\ \forall i\in[m]$ and $\sum_{i=1}^{m}|c_{i}|^{2}=1$, and if we measure this quantum state, we get $f(x_{i})+\xi$ with probability $|c_{i}|^{2}$, and $\xi$ follows a sub-Gaussian distribution as in Eq. (2). We will add more discussions in intro or Section 2.1. Regarding $\xi$, it was a typo, which should be a scalar in $\mathbb{R}$ (similar to the fact that the $\epsilon_x$ in Eq. (2) is a scalar); we thank the reviewer for spotting this and will correct this in future versions.

---

### Official Review · Reviewer_RonT · 2022-07-12

**Rating:** 6
**Confidence:** 2
**Soundness:** 3 good
**Presentation:** 2 fair
**Contribution:** 3 good

**Summary:**

Main Contributions:
1. The first study of Approximately convex and stochastic convex functions using Quantum Algorithms
2. The application of these algorithms and performance bounds to bandit algorithms resulting in a logarithmic regret

The authors use a more general problem of approximately- and stochastic- convex optimisation using Quantum algorithms to then derive specific results for the Bandit Algorithm Case.

**Questions:**

Questions & Suggestions:
1. What does poly(e, 1/n) mean, as there are two arguments. [Line 32]
2. Zeroth-Order Bandit Algorithms / Oracles are a confusing terminology and a reference, footnote, or detailed explanation would be helpful.


**Limitations:**

Limitations:
1. The claim of logarithmic bounds is only true due to an associated probability of achieving the result, hence direct comparison is not possible. A discussion of this would be helpful.
2. How realistic is this algorithm in terms of Qubits required to run (given what problem dimension). Discussion of this would be interesting.

Societal Impact:
1. Everything seems to have been discussed.

**Strengths And Weaknesses:**

Strenghts:
1. Clever use of stochastic optimisation in general leads to a direct application to Bandit algorithms yielding a logarithmic regret bound.

Weaknesses:
1. Unclear if you are proposing the first Quantum algorithm that has the Logarithmic Quantum regret bound, if so then it is very big, if not (which is more likely) then a comparison against existing methods is missing.
2. The Theorems 2 & 3 suggest seemingly arbitrary probablity bounds 0.9 and 0.8 resp. this should have more explanation.
3. (ideally) Theorems 2 & 3 would have a delta (representing the probability) with which the result can be found (and it will then also feature in the formula) as opposed to fixed probabilites 0.9 and 0.8.

---

> ### Author Response · Authors · 2022-07-29
> **Official Reply**
>
> We thank the reviewer for the detailed comments.
>
> As far as we know, our quantum algorithm is the first one with poly-logarithmic quantum regret bound among previous literature on online optimization algorithms by the time we made this NeurIPS submission. After the NeurIPS submission deadline, we became aware that an independent work https://arxiv.org/abs/2205.14988 gave a quantum algorithm with poly-logarithmic regret for multi-armed bandits and stochastic linear bandits. In the future version of our paper, we will emphasize that we give the first quantum algorithms with poly-logarithmic regret, and also make a comparison to arXiv:2205.14988.
>
> The 0.9 and 0.8 in our Theorem 1 and Corollary 1 can be replaced by 1-delta, and there will be an overhead of log(1/delta) in the quantum query complexities. We will fix the theorem statement and proofs in future versions. We believe that making such changes will also resolve the concern that the claim of logarithmic bounds is only true due to an associated probability. Nevertheless, we would like to point out that in the current version, our poly-logarithmic regret bound in Theorem 2 is a result in expectation, and can be directly comparable to classical state-of-the-arts such as Theorem 1 of Ref. [23].
>
> The poly(n,1/eps) in Line 32 represents a formula with polynomial dependence in both n (the dimension) and 1/eps (where epsilon is an error parameter in Eq. (1)).
>
> Zeroth-Order Bandit Algorithms / Oracles have common appearances in existing literature, for instance Hazan and Li (https://arxiv.org/pdf/1603.04350.pdf), Lattimore (https://arxiv.org/pdf/2006.00475.pdf), Lattimore and Gyorgy (Ref. [23]), etc. We will add these references for clarification when we first mention zeroth-order bandits.
>
> It is a nice question about the number of qubits required to run our quantum algorithm. For each state $|\pi\rangle$, we use $\widetilde{O}(n)$ qubits to store it. Then, in the quantum walk process, we use another $O(n)$ ancilla qubits to implement the quantum walk operators (U and R). In the whole algorithm, we use $\widetilde{O}(n)$ copies of the state $|\pi\rangle$. Hence, the total number of logical qubits is $\widetilde{O}(n^2)$. However, to run our algorithm in a real quantum device, more physical qubits are needed for quantum error correction.

---

### Official Review · Reviewer_r2eB · 2022-07-26

**Rating:** 7
**Confidence:** 3
**Soundness:** 4 excellent
**Presentation:** 3 good
**Contribution:** 3 good

**Summary:**

The manuscript "Quantum Speedups of Optimizing Approximately Convex Functions with Applications to Logarithmic Regret Stochastic Convex Bandits" investigates the use of quantum algorithms for the optimization of approximately convex functions. In particular, the authors describe how to achieve a polynomial quantum speedup in terms of the dimension $n$ compared to the best-known classical method by exploiting quantum walks. The framework is then adapted to give a quantum algorithm for zeroth-order stochastic convex bandits, showing a exponential speedup in the number of rounds $T$.


**Questions:**

(1) page 2, line 43: missing comma after "technology"

(2) page 2, line 54: It is clear what is meant by "... if we measure *this* quantum state, ...", but the formulation is confusing.

(3) page 2, lines 61-62: refer to Appendix C for details and proof

(4) page 5, line 177: Why is the integration variable changed in the definition of $\psi_x$? On page 4, line 143 and page 13, line 439, the integral is defined w.r.t. the second variable of $P$. This is also the case in [34,35].

(5) page 5, line 183: "require*s*"

(6) page 6, line 186: Why the term "user-fiendly"? Could you elaborate on that?

(7) page 7, line 244: Instead of having two nearly identical pseudo codes, it might make sense to replace Alg. 1 by Alg. 6 without repetition in Appendix C.3.

(8) page 9, line 283: $\mathcal{O}^{stoc}_f = \mathcal{O}^{\tau,\alpha}_f$ in Alg. 2, line 12?

(9) page 11, lines 380-383: [35] and [36] are identical.

(10) pages 13-21: Three different notations for the transition probabilities are used in the Appendix: $P(x,y)$ and $p_{x \to y}$ on page 13 and $P_{x,y}$ on page 21. The first variant is also used in Sections 2 and 3, so it would be coherent to stick to that.

(11) page 13, line 450: $D$ is not defined.

(12) page 14, lines 460 and 469: The order of integration is not correct.

(13) page 14, line 473: "Theroem"

(14) page 15, lines 516-522: Is Theorem 3 the informal version of Corollary 2?

(15) page 18, Alg. 5, lines 7 and 8: $\Sigma^\prime_i = \Sigma^\prime$?

(16) page 21, lines 632/633: One $\ket{x}$ too much on the right-hand side of the equation.

(17) page 21, lines 637-640: A bit more detail on the last steps of the implementation of the quantum walk operator would be helpful.

**Limitations:**

Yes, the authors adequately addressed the limitations and potential negative societal impact of their work.

**Strengths And Weaknesses:**

The presented work is purely theoretical. The authors start with clear statements of their contributions in form of two major theorems and address open questions. Then, the basic concepts of quantum computing, classical and quantum walks, and Hit-and-Run walks are explained in a clear and comprehensible fashion. Section 3 gives a condensed overview of the main results and their derivations. That is, a "user-friendly" quantum walk framework is introduced and the authors show how to apply this framework for optimizing approximately convex functions. Both parts are reconsidered and described in detail in the Appendix. This way of structuring the paper is a good idea, but leads to some repeated or redundant statements.

Overall, the manuscript is well-written and all algorithms and results are described in greater detail. The work is very interesting and shows that the use of quantum walks can improve the query complexity of sampling methods. I recommend it for publication after minor revision

---

> ### Author Response · Authors · 2022-07-29
> **Official Reply**
>
> We thank the reviewer for the general positive opinions about our paper, and the very detailed suggestions! We will adopt them in future versions of our paper.

---

### Official Review · Reviewer_Dqks · 2022-07-26

**Rating:** 5
**Confidence:** 2
**Soundness:** 3 good
**Presentation:** 3 good
**Contribution:** 2 fair

**Summary:**

This work considers the task of minimizing approximately convex functions and that of regret minimization for stochastic convex bandits. The results here using a quantum evaluation oracle offers an improvement (in T for bandits, and dimension for optimization) compared to classical models of access.


**Questions:**

Please see above. Additionally—

— Can the authors comment on the number of calls made to the membership oracle for the set K? Quantum algorithms also have an advantage in this sense because separation oracles carry the same complexity as membership ones (e.g. https://arxiv.org/abs/1809.00643)


**Limitations:**

Yes

**Strengths And Weaknesses:**

For someone unfamiliar with quantum algorithms (like me), the first disclaimer here is that these speed-ups result from the being able to query the noisy function in superposition. These are stronger models of access than classical counterparts, and can therefore overcome classical lower bounds like T^0.5 for regret.

The optimization result is easier to understand of the two. The speed-ups result from amplitude amplification (of which Grover is an example) like properties of quantum hit-and-run. The algorithmic primitives that avoid decoherence seem sound.

I am concerned regarding the regret result. The minimization of approximately convex immediately yields an improvement (n^5/eps) for sample complexity of the best point with access to noisy evaluations. This is sound and this reduction is present in BLNR; here there’s an added advantage of being able to estimate the mean of a Gaussian at 1/N rate.

The paper claims a regret result. But to the best of my reading, the regret is not measured on the points the algorithm actually queries (which are quantum superpositions), but an a separate (vector) point chosen as “output”. If so, this goes against the usual definition, where exploration induces a penalty absent here. Can the authors comment on this? Failing this, I think it would be best to present this as a best-arm result (which is also what BLNR does).

---

> ### Author Response · Authors · 2022-07-29
> **Official Reply**
>
> We thank the reviewer’s great effort and valuable comments.
>
> The question about our regret definition is a fundamental one and we are happy to make more comments on this. The main difference between quantum and classical here is that quantumly we query points in superposition, which does not give a definite output and hence cannot directly apply the classical definition of regrets. As a result, it is pretty tricky to define the notion of regret in the quantum setting properly, and to characterize the regret cumulatively in all iterations, our setting is to assign a point (based on previous quantum queries) and use this in the definition of quantum regret. A closely-related paper was an independent one released after the NeurIPS 2022 submission deadline which gives poly-logarithmic regret quantum algorithms for multi-armed bandits and stochastic linear bandits https://arxiv.org/pdf/2205.14988.pdf (see also our reply to Reviewer RonT), but in the bandit setting they do not have superposition between different arms, which is a fundamental difference between our work and their work. As far as we know, our paper is the first work initiating the study of quantum regrets of online learning problems, and we look forward to future works that can propose other models and extend corresponding studies.
>
> For the membership oracle, in the current version of this paper, we consider K as a known space, like R^n or sphere S^n, where we can do sampling without querying an oracle. However, it is a very nice question to consider an unknown space K.
>
> - In our algorithm, the first place we need access to K is preparing $N=\widetilde{O}(n)$ copies of the initial state $|\pi_0\rangle$, which corresponds to a uniform superposition of the points in K. Each state can be prepared by querying the membership oracle for$ \widetilde{O}(1)$ times. Thus, in total, it queries the membership oracle for $\widetilde{O}(n)$ times.
> - Another place we access K is in each step of the Hit-and-Run walk, where we can use $\widetilde{O}(1)$ queries to the membership oracle to determine the two intersect points of l \cap K. Thus, the total number of membership oracle queries is of the same order as the number of evaluation oracle queries. Thus, in total, it queries the membership oracle for $\widetilde{O}(n^3)$.
>
> Therefore, the total number of the membership oracle queries is $\widetilde{O}(n^3)$. For the separation oracle, according to the results in van Apeldoorn et al. (Ref. [5]) and Chakrabarti et al. (Ref. [15]), it is $\widetilde{O}(1)$-equivalent to the membership oracle in the quantum setting. Thus, if we have access to a separation oracle, the query complexity is still $\widetilde{O}(n^3)$. We thank again for this very helpful comment and will add this discussion in the final version of this paper.

---

> > ### Comment · Reviewer_Dqks · 2022-08-09
> > **Update**
> >
> > Thanks for the response.
> >
> > I still disagree with the term 'regret', and still think 'best-arm' is the appropriate categorization. However, I also think it is counter-productive for the discussion to get stuck on naming/terminology. This disagreement is not reflected in my score.
> >
> > Given a query $\int_x \sqrt{p(x)} \vert x\rangle dx$, a natural choice to measure regret is evaluate the leaner's loss as $\int_x p(x) f(x) dx$. Evaluation the performance on a different arm than the one queried is a defining feature of best-arm problems.

---

### Meta-Review · Area_Chair_V71m · 2022-08-21

**Recommendation:** Accept
**Confidence:** Less certain

**Metareview:**

Interesting paper - a little bit on the weak side in terms of presentation especially given the community. The consensus is around a weak accept and I concur.

**Award:**

No

---

### Decision · Program_Chairs · 2022-09-14

Accept